# Provable Robust Learning for Deep Neural Networks under Agnostic Corrupted Supervision

## Abstract

Training deep neural models in the presence of corrupted supervisions is challenging as the corrupted data points may significantly impact the generalization performance. To alleviate this problem, we present an efficient robust algorithm that achieves strong guarantees without any assumption on the type of corruption, and provides a unified framework for both classification and regression problems. Different from many existing approaches that quantify the quality of individual data points (e.g., loss values) and filter out data points accordingly, the proposed algorithm focuses on controlling the collective impact of data points on the averaged gradient. Even when a corrupted data point failed to be excluded by the proposed algorithm, the data point will have very limited impact on the overall loss, as compared with state-of-the-art filtering data points based on loss values. Extensive empirical results on multiple benchmark datasets have demonstrated the robustness of the proposed method under different types of corruptions.

## 1 Introduction

Corrupted supervision is a common issue in real-world learning tasks, where the learning targets are not accurate due to various factors in the data collection process. In deep learning models, such corruptions are especially severe, whose degree-of-freedom makes them easily memorize corrected examples and susceptible to overfitting (Zhang et al., 2016).

There are extensive efforts to achieve robustness against corrupted supervisions. A natural approach to deal with corrupted supervision in deep neural networks (DNNs) is to reduce the model exposure to corrupted data points during training. By detecting and filtering (or re-weighting) the possible corrupted samples, the learning is expected to deliver a model that is similar to the one trained on clean data (without corruption) (Kumar et al., 2010; Han et al., 2018; Zheng et al., 2020). There are different criteria designed to identify the corrupted data points in training. For example, Kumar et al. (2010); Han et al. (2018); Jiang et al. (2018) leveraged the loss function values of data points; Zheng et al. (2020) tapped prediction uncertainty for filtering data; Malach & Shalev-Shwartz (2017) used the disagreement between two deep networks; Reed et al. (2014) utilized the prediction consistency of neighboring iterations. The success of these methods highly depends on the effectiveness of the detection criteria in correctly identifying the corrupted data points. Since the corrupted labels remain unknown throughout the learning, such "unsupervised" detection approaches may not be effective, either lack theoretical guarantees of robustness (Han et al., 2018; Reed et al., 2014; Malach & Shalev-Shwartz, 2017; Li et al., 2017) or provide guarantees under assumptions of the availability of prior knowledge about the type of corruption (Zheng et al., 2020; Shah et al., 2020; Patrini et al., 2017; Yi & Wu, 2019). Besides, another limitation of many existing approaches is that, they are exclusively designed for classification problems (e.g., Malach & Shalev-Shwartz (2017); Reed et al. (2014); Menon et al. (2019); Zheng et al. (2020)) and are not straightforward to extend to solve regression problems.

To tackle these challenges, this paper presents a unified optimization framework with robustness guarantees without any assumptions on how supervisions are corrupted, and is applicable to both classification and regression problems. Instead of developing an accurate criterion for detection corrupted samples, we adopt a novel perspective and focus on limiting the collective impact of corrupted samples during the learning process through *robust mean estimation* of gradients. Specifically, if our estimated average gradient is close to the gradient from the clean data during the learning iterations,

then the final model will be close to the model trained on clean data. As such, a corrupted data point can still be used during the training when it does not considerably alter the averaged gradient. This observation has remarkably impact on our algorithm design: instead of explicitly quantifying (and identifying) individual corrupted data points, which is a hard problem in itself, we are now dealing with an easier task, i.e., eliminating training data points that significantly distort the mean gradient estimation. One immediate consequence of this design is that, even when a corrupted data point failed to be excluded by the proposed algorithm, the data point is likely to have very limited impact on the overall loss, as compared with state-of-the-art filtering data points based on loss values. We perform experiments on both regression and classification with corrupted supervision on multiple benchmark datasets. The results show that the proposed method outperforms state-of-the-art.

## 2 BACKGROUND

Learning from corrupted data (Huber, 1992) has attracted considerable attention in the machine learning community (Natarajan et al., 2013). Many recent studies have investigated robustness of classification tasks with noisy labels. For example, Kumar et al. (2010) proposed a self-paced learning (SPL) approach, which assigns higher weights to examples with smaller loss. A similar idea was used in curriculum learning (Bengio et al., 2009), in which the model learns easy samples first before learning harder ones. Alternative methods inspired by SPL include learning the data weights (Jiang et al., 2018) and collaborative learning (Han et al., 2018; Yu et al., 2019). Label correction (Patrini et al., 2017; Li et al., 2017; Yi & Wu, 2019) is another approach, which revises original labels in data with a goal to recover clean labels from corrupt ones. However, since we do not have access to which data points are corrupted, it is hard to get provable guarantees for label correction without strong assumptions regarding the corruption type.

Accurate estimation of gradients is a key step for successful optimization. The relationship between gradient estimation and its final convergence has been widely studied in the optimization community. Since computing an approximated (and potentially biased) gradient is often more efficient than computing the exact gradient, many studies used approximated gradients to optimize their models and showed that they suffer from the biased estimation problem if there is no assumptions on the gradient estimation (d'Aspremont, 2008; Schmidt et al., 2011; Bernstein et al., 2018; Hu et al., 2020; Ajalloeian & Stich, 2020).

A closely related topic is robust estimation of the mean. Given corrupted data, robust mean estimation aims at generating an estimated mean $\hat{\mu}$ such that the difference between the estimated mean on corrupted data and the mean of clean data $\|\hat{\mu} - \mu\|_2$ is minimized. It was showed that median or trimmed-mean are the optimal statistics for mean estimation in one-dimensional data (Huber, 1992). However, robustness in high dimension is quite challenging since applying the coordinate-wise optimal robust estimator would lead to an error factor $\mathcal{O}(\sqrt{d})$ that scales with the data dimension. Although some classical work, such as Tukey median (Tukey, 1975), successfully designed algorithms to get rid of the $\mathcal{O}(\sqrt{d})$ error, the algorithms themselves are not polynomial-time algorithm. More recently, Diakonikolas et al. (2016); Lai et al. (2016) successfully designed polynomial-time algorithms with dimension-free error bounds. The results have been widely applied to improve algorithmic efficiency in various scenarios (Dong et al., 2019; Cheng et al., 2020).

Robust optimization aims to optimize the model given corrupted data. Many previous studies improve the robustness of the optimization in different problem settings. However, most of them either study linear regression and its variantes(Bhatia et al., 2015; 2017; Shen & Sanghavi, 2019) or study the convex optimization (Prasad et al., 2018). Thus, those results cannot be directly generalized to deep neural networks. Diakonikolas et al. (2019) is a very generalized non-convex optimization method with the agnostic corruption guarantee. However, the space complexity of the algorithm is high, thus cannot be applied to deep neural networks given current hardware limitations.

## 3 METHODOLOGY

Before introducing our algorithm, we first discuss the corrupted supervision. To characterize agnostic corruptions, we make use of an *adversary* that tries to corrupt the supervision of a clean data. There is no limitation on how the adversary corrupts the supervision, which can either be randomly permuting the target, or in a way that maximizes the negative impact (i.e., lower performance).

Firstly, the adversary can choose up to $\epsilon$ fraction of the clean target $\mathbf{D}_y \in \mathbb{R}^{n \times q}$ and change the selected row of $\mathbf{D}_y$ to arbitrary valid numbers, generating $\mathbf{D}_y^\epsilon \in \mathbb{R}^{n \times q}$. Then, the adversary returns the corrupted dataset $\mathbf{D}_x, \mathbf{D}_y^\epsilon$ to our learning algorithm $\mathcal{A}$. In this process, the only constraint on the adversary is the $\epsilon$ fraction, and the adversary has full knowledge of the data, and even the learning algorithm $\mathcal{A}$. A natural question to ask is: *Given a data set with $\epsilon$-fraction corrupted supervision* $\mathbf{D}_x \in \mathbb{R}^{n \times p}$, $\mathbf{D}_y^\epsilon$, *and a learning objective* $\phi : \mathbb{R}^p \times \mathbb{R}^q \times \mathbb{R}^d \to \mathbb{R}$ *parameterized by* $\theta$, *can we output parameters* $\theta \in \mathbb{R}^d$ *such that* $\|\nabla_\theta \phi(\theta; \mathbf{D}_x, \mathbf{D}_y)\|$ *is minimized.*

When $\epsilon = 0$, we have $\mathbf{D}_y^\epsilon = \mathbf{D}_y$ and learning is done on the clean data. The stochastic gradient descent could converge to a stationary point, where $\|\nabla_\theta \phi(\theta; \mathbf{D}_x, \mathbf{D}_y)\| = 0$. However, when the supervision is corrupted as above, this is not the case any more, due to the error in $\theta$ impacted by the corrupted data. We thus want an efficient algorithm to find a model $\theta$ that minimizes $\|\nabla_\theta \phi(\theta; \mathbf{D}_x, \mathbf{D}_y)\|$. A robust model $\theta$ should have a small value of $\|\nabla_\theta \phi(\theta; \mathbf{D}_x, \mathbf{D}_y)\|$, and we hypothesize that a smaller $\|\nabla_\theta \phi(\theta; \mathbf{D}_x, \mathbf{D}_y)\|$ has better generalization.

## 3.1 STOCHASTIC GRADIENT DESCENT WITH BIASED GRADIENT

A direct consequence of corrupted supervision is biased gradient estimation. In this section, we will first analyze how such biased gradient estimation affects the robustness of learning. The classical analysis of stochastic gradient descent (SGD) requires access to the stochastic gradient oracle, which is an unbiased estimation of the true gradient. However, corrupted supervision leads to corrupted gradients, and it is thus difficult to get unbiased gradient estimation without assumptions of how the gradients are corrupted. We start the analysis by the following informal theorem (without elaborated discussions of assumptions) of how biased gradient affects the final convergence of SGD. Its formal version is provided in Theorem 4, Appendix.

**Theorem 1 (Convergence of Biased SGD (Informal))** *Under mild assumptions, denote $\zeta$ to be the maximum $\ell_2$ norm of the difference between clean minibatch gradient and corrupted minibatch gradient $\|\mathbf{g} - \tilde{\mathbf{g}}\| \leq \zeta$, then by using biased gradient estimation, SGD converges to the $\zeta$-approximated stationary points:* $\mathbb{E}\|\nabla \phi(\theta_t)\|^2 = \mathcal{O}(\zeta^2)$.

**Remark 1** *In the corrupted supervision setting, let the gradient estimated by corrupted data $\mathcal{D}_\epsilon$ be $\hat{\mathbf{g}}$, the gradient estimated by clean data $\mathcal{D}$ be $\mathbf{g}$. Assume $\|\tilde{\mathbf{g}} - \mathbf{g}\| \leq \zeta$, it follows that when using corrupted dataset in SGD, it converges to the $\zeta$-approximated stationary point of the objective defined by the clean data. Note the difference between above theorem and typical convergence theorem is that we are using a biased gradient estimation.*

According to Theorem 1 and the remark, a robust estimation of the gradient $\mathbf{g}$ is the key to ensure a robust model (converge to the clean solution). We also assume the loss function has the form of $\mathcal{L}(\mathbf{y}, \hat{\mathbf{y}})$, where many commonly used loss functions fall in this category.

## 3.2 ROBUST GRADIENT ESTIMATION FOR GENERAL DATA CORRUPTION

We first introduce Algo. 2 for general corruption (i.e. corruption on both features and/or supervisions). The algorithm excludes the data points with large gradient norms, and uses the empirical mean of the remaining to update gradients. In Thm. 2 we give its robustness property.

---

**Algorithm 1:** Robust Mean Estimation for Corrupted Gradient

**input:** gradient matrix $\mathbf{G} \in m \times d$, corruption rate $\epsilon$
**return** *estimated mean* $\hat{\mu} \in \mathbb{R}^d$ ;
1. For each row $\mathbf{z}_i$ in $\mathbf{G}$, calculate the l2 norm $\|\mathbf{z}_i\|$
2. Choose the $\epsilon$-fraction rows with large $\|\mathbf{z}_i\|$
3. Remove those selected rows, and return the empirical mean of the rest points as $\hat{\mu}$.

---

**Assumption 1 (Individual $L$-smooth loss)** *For every individual loss function $\phi_i$, there exists constant $L > 0$, such that for a clean sample $i$, we have $|\phi_i(\mathbf{x}) - \phi_i(\mathbf{y})| \leq L|\mathbf{x} - \mathbf{y}|$ for any $\mathbf{x}, \mathbf{y}$.*

**Theorem 2 (Robust Gradient Estimation For Data Corruption)** *Let $\tilde{\mathbf{G}} \in \mathbb{R}^{m \times d}$ be a corrupted gradient matrix, and $\mathbf{G} \in \mathbb{R}^{m \times d}$ be the clean gradient matrix. Let $\mu$ be the empirical mean function,*

---

**Algorithm 2: (*PRL(G)*) Provable Robust Learning for General Corrupted Data**

---

**input:** Label corrupted dataset $\mathbf{D}_x, \mathbf{D}_y^\epsilon$, learning rate $\gamma_t$;
**return** *model parameter* $\theta$;
**for** *t = 1 to maxiter* **do**
    Randomly sample a minibatch $\mathbf{M}$ from $\mathbf{D}_x, \mathbf{D}_y^\epsilon$
    Calculate the individual gradient $\tilde{\mathbf{G}}$ for $\mathbf{M}$
    Apply Algorithm 1 on $\tilde{\mathbf{G}}$ to get robust gradient estimation $\hat{\mu}$
    Update model $\theta_{t+1} = \theta_t - \gamma_t \hat{\mu}$
**end**

---

*we have that the output of Algo. 1 $\hat{\mu}$ of $\tilde{\mathbf{G}}$ satisfies $\|\mu(\mathbf{G}) - \hat{\mu}\| = \mathcal{O}(\epsilon\sqrt{d})$. Moreover, if Asm. 1 holds, we further have $\|\mu(\mathbf{G}) - \hat{\mu}\| = \mathcal{O}(\epsilon L)$.*

Combining with the aforementioned convergence analysis of biased SGD, we get the following:

**Corollary 1 (Robust Optimization For Corrupted Data)** *Given assumptions used in Thm. 1, and Asm. 1, applying Algo. 1 to any $\epsilon$-fraction corrupted data, we get $\min_{t \in [T]} \mathbb{E}\|\nabla\phi(\mathbf{x}_t)\| = \mathcal{O}(\epsilon L)$ with large enough $T$. If Asm. 1 does not hold, then we get $\min_{t \in [T]} \mathbb{E}\|\nabla\phi(\mathbf{x}_t)\| = \mathcal{O}(\epsilon\sqrt{d})$ with large enough $T$.*

The robustness guarantee states that even training on generally corrupted data (corrupted supervision is a special case), Algo. 2 guarantee that the gradient norm on remaining data cannot be too large. Since Thm. 2 gives a dimension-free error bound when Asm. 1 holds, Corollary 1 also gives the dimension-free robustness guarantee with Asm. 1. We defer the detailed discussion of $\mathcal{O}(\epsilon L)$ to later sections. Although the error bound $\mathcal{O}(\epsilon L)$ sounds good, we note that it still has several drawbacks:

First, the dimension-free error bound means the error does not grow with increasing dimensions, and is critical when working with neural networks, due to the extremely large gradient dimension (i.e., #parameters of neural network). Thm. 2 gives the dimension-free error bound only when Asm. 1 holds, which is quite strong. In addition, even when Asm. 1 holds, $L$ can be large, leading to a large gradient estimation error. Existing work (Diakonikolas et al., 2019) already acheives the dimension-free $\mathcal{O}(\sqrt{\epsilon})$ guarantee with general corruptions, which is a much more better theoretical results than above theorem. However, in practice, we found that the gradient norms of deep neural networks for individual data points are usually not very large, even at the beginning of the training. This can be partially due to the network structure. Further discussion on this issue is beyond the scope of this paper, but the theoretical bound above states that the robustness should depend on the number of parameters for the general models.

Another concern of Alg. 2 is the efficiency. It requires computing individual gradients. Although there are some advanced approaches to get the individual gradient, e.g., (Goodfellow, 2015), it is still relatively slow as compared to commonly used back-propagation. Moreover, these methods are usually not compatible with popular components such as batch normalization (BN) since the individual gradients are not independent inside BN, using of which will lose the benefits from parallelization.

### 3.3 ROBUST GRADIENT ESTIMATION FOR ONE DIMENSIONAL CORRUPTED SUPERVISION

In this section, we show that the above robustness bound can be improved if we assume the corruption only comes from supervision. Also, by fully exploiting the gradient structure of the corrupted supervision, our algorithm can be much more efficient and meanwhile compatible with batch normalization. We use the one dimensional supervision setting (binary classification or single-target regression) to illustrate this intuition and extend it more general settings in the next section. Consider a high-dimensional supervised learning problem with $\mathbf{X} \in \mathbb{R}^{n \times p}$ and $\mathbf{y} \in \mathbb{R}^n$. The goal is to learn a function $f$ parameterized by $\theta \in \mathbb{R}^d$ minimizing the following loss $\min_\theta \sum_{i=1}^n \phi_i = \min_\theta \sum_{i=1}^n \mathcal{L}(y_i, f(\mathbf{x}_i, \theta))$. The gradient for a data point $i$ is $\nabla_\theta \phi_i = \frac{\partial l_i}{\partial f_i} \frac{\partial f_i}{\partial \theta} = \alpha_i \mathbf{g}_i$.

One key observation is that: when only supervision is corrupted, then the corruption contributes only to the term $\alpha_i = \frac{\partial l_i}{\partial f_i}$, which is a scalar in the one-dimensional setting. In other words, given the clean gradient of $i^{th}$ point, $g_i \in \mathbb{R}^d$, the corrupted supervision can only perturbs the the length of the gradient vector, changing the gradient from $\alpha_i \mathbf{g}_i$ to $\delta_i \mathbf{g}_i$, where $\delta_i = \frac{\partial l_i^\epsilon}{\partial f_i}$. When $\alpha_i$ and $\delta_i$ are

known, then we can easily eliminate the impact from corrupted supervision. But this is not the case since we have have only the possibly corrupted target $\hat{y}_i$ as opposed to the ground truth $y_i$.

On the other hand, the fact that corrupted supervision scales the clean gradient can be used to reshape the robust optimization problem. Recall that in every iteration, we update our model by $\theta^+ = \theta - \gamma\mu(\mathbf{G})$, where $\mu$ denotes the empirical mean function and $\mathbf{G} = [\nabla_\theta\phi_1^T, \ldots, \nabla_\theta\phi_m^T] \in \mathbb{R}^{m \times d}$ is the gradient matrix with mini-batch size $m$. We then have the following:

**Problem 1 (Robust Gradient Estimation for Corrupted Supervision - One Dimensional Case)**
*Given a clean gradient matrix $\mathbf{G} \in \mathbb{R}^{m \times d}$, an $\epsilon$-corrupted matrix $\tilde{\mathbf{G}}$ with at most $\epsilon$-fraction rows are corrupted from $\alpha_i\mathbf{g}_i$ to $\delta_i\mathbf{g}_i$, design an algorithm $\mathcal{A} : \mathbb{R}^{m \times d} \to \mathbb{R}^d$ that minimizes $\|\mu(\mathbf{G}) - \mathcal{A}(\tilde{\mathbf{G}})\|$.*

Note that when $\|\delta_i\|$ is large, the corrupted gradient will have large effect on the empirical mean, and vice versa. This motivates us to develop an algorithm that filters out data points by the loss layer gradient $\|\frac{\partial l_i}{\partial f_i}\|$. If the norm of the loss layer gradient of a data point is large (in one-dimensional case, this gradient reduces to a scalar and the norm becomes its absolute value), we exclude the data point when computing the empirical mean of gradients for this iteration. Note that this algorithm is applicable to both regression and classification problems. Especially, when using the mean squared error (MSE) loss for regression, its gradient norm is exactly the loss itself, and the algorithm reduces to self-paced learning Kumar et al. (2010). We summarize the procedure in Alg. 3 and extend it to the more general multi-dimension case in the next section.

---

**Algorithm 3: (*PRL(L)*) Efficient Provable Robust Learning for Corrupted Supervision**

---

**input:** dataset $\mathbf{D}_x, \mathbf{D}_y^\epsilon$ with corrupted supervision, learning rate $\gamma_t$;
**return** *model parameter $\theta$*;
**for** *t = 1 to maxiter* **do**
    Randomly sample a minibatch $\mathbf{M}$ from $\mathbf{D}_x, \mathbf{D}_y^\epsilon$
    Compute the predicted label $\hat{\mathbf{Y}}$ from $\mathbf{M}$
    Calculate the gradient norm for the loss layer, (i.e. $\|\hat{\mathbf{y}} - \mathbf{y}\|$ for mean square error or cross entropy) for each data point in $\mathbf{M}$
    Remove the top $\tau$-fraction data from $\mathbf{M}$ according to $\|\hat{\mathbf{y}} - \mathbf{y}\|$
    Return the empirical mean of the remaining $\mathbf{M}$ as the robust mean estimation $\hat{\mu}$
    Update model $\theta_{t+1} = \theta_t - \gamma_t\hat{\mu}$
**end**

---

### 3.4 Extension to Multi-Dimensional Corrupted Supervision

To extend our algorithm and analysis to multi-dimensional case, let $q$ to be the supervision dimension, the gradient for each data point is $\nabla_\theta\phi_i = \frac{\partial l_i}{\partial f_i}\frac{\partial f_i}{\partial \theta}$, where $\frac{\partial l_i}{\partial f_i} \in \mathbb{R}^q$ is the gradient of loss respect to model outputs, and $\frac{\partial f_i}{\partial \theta} \in \mathbb{R}^{q \times d}$ is the gradient of model outputs respect to model parameters. Similarly, when the supervision is corrupted, the corruption comes from the term $\frac{\partial l_i}{\partial f_i}$, which is a vector. Let $\delta_i = \frac{\partial l_i^\epsilon}{\partial f_i} \in \mathbb{R}^q$, $\alpha_i = \frac{\partial l_i}{\partial f_i} \in \mathbb{R}^q$, $\mathbf{W}_i = \frac{\partial f_i}{\partial \theta} \in \mathbb{R}^{q \times d}$, $m$ be the minibatch size. Denote the clean gradient matrix $\mathbf{G} \in \mathbb{R}^{m \times d}$, where the $i_{th}$ row of gradient matrix $\mathbf{g}_i = \alpha_i\mathbf{W}_i$. Now the multi-dimensional robust gradient estimation problem is defined by:

**Problem 2 (Robust Gradient Estimation for Corrupted Supervision - Multi-Dimensional Case)**
*Given a clean gradient matrix $\mathbf{G}$, an $\epsilon$-corrupted matrix $\tilde{\mathbf{G}}$ with at most $\epsilon$-fraction rows are corrupted from $\alpha_i\mathbf{W}_i$ to $\delta_i\mathbf{W}_i$, design an algorithm $\mathcal{A} : \mathbb{R}^{m \times d} \to \mathbb{R}^d$ that minimizes $\|\mu(\mathbf{G}) - \mathcal{A}(\tilde{\mathbf{G}})\|$.*

We start our analysis by investigating the effects of the filtering-base algorithm, i.e. use the empirical mean gradient of $(1 - \epsilon)$-fraction subset to estimate the empirical mean gradient of clean gradient matrix. We have the following for *a randomized filtering-based algorithm*(proof in Appendix):

**Lemma 1 (Gradient Estimation Error for Random Dropping $\epsilon$-fraction Data)** *Let $\tilde{\mathbf{G}} \in \mathbb{R}^{m \times d}$ be a corrupted matrix generated as in Problem 2, and $\mathbf{G} \in \mathbb{R}^{m \times d}$ be the original clean gradient matrix. Suppose arbitrary $(1 - \epsilon)$-fraction rows are selected from $\tilde{\mathbf{G}}$ to form the matrix $\mathbf{N} \in \mathbb{R}^{n \times d}$. Let $\mu$ be the empirical mean function. Assume the clean gradient before loss layer has bounded*

*operator norm, i.e., $\|\mathbf{W}\|_{op} \leq C$, the maximum clean gradient in loss layer $\max_{i \in \mathbf{G}} \|\alpha_i\| = k$, the maximum corrupted gradient in loss layer $\max_{i \in \mathbf{N}} \|\delta_i\| = v$, then we have:*

$$\|\mu(\mathbf{G}) - \mu(\mathbf{N})\| \leq Ck\frac{3\epsilon - 4\epsilon^2}{1 - \epsilon} + Cv\frac{\epsilon}{1 - \epsilon}.$$

We see that $v$ is the only term that is related to the corrupted supervision. If $v$ is large, then the bound is not safe since the right-hand side can be arbitrarily large (i.e. an adversary can change the label in a way such that $v$ is extremely large). Thus controlling the magnitude of $v$ provides a way to effectively reduce the bound. For example, if we manage to control $v \leq k$, then the bound is safe. This can be achieved by sorting the gradient norms at the loss layer, and then discarding the largest $\epsilon$-fraction data points. We thus have the following result.

**Theorem 3 (Robust Gradient Estimation For Supervision Corruption)** *Let $\tilde{\mathbf{G}}$ be a corrupted matrix generated in Problem 2, $q$ be the label dimension, $\mu$ be the empirical mean of clean matrix $\mathbf{G}$. Assume the maximum clean gradient before loss layer has bounded operator norm: $\|\mathbf{W}\|_{op} \leq C$, then the output of gradient estimation in Algo 3 $\hat{\mu}$ satisfies $\|\mu - \hat{\mu}\| = \mathcal{O}(\epsilon\sqrt{q}) \approx \mathcal{O}(\epsilon)$.*

Compare Thm. 2 and Thm. 3, we see that when the corruption only comes from supervision, the dependence on $d$ is reduced to $q$, where in most deep learning cases we have $d \gg n$. Applying Thm 1 directly shows that our algorithm is also robust in multi-label settings.

### 3.5 COMPARISON WITH DIAKONIKOLAS ET AL. (2019) AND OTHER METHODS

SEVER (Diakonikolas et al., 2019) showed promising state-of-the-art theoretical results in general corruptions, which achieves $\mathcal{O}(\sqrt{\epsilon})$ dimension-free guarantee for general corruptions. Compared to Diakonikolas et al. (2019), we have two contributions: **a)**. By assuming the corruption comes from the label (we admit that this is quite strong compared to the general corruption setting), we could get a better error rate. **b)**. Our algorithm can be scaled to deep neural networks while Diakonikolas et al. (2019) cannot. We think this is a contribution considering the DNN based models are currently state-of-the-art methods for noisy label learning problems (at least in empirical performance).

Although Diakonikolas et al. (2019) achieves very nice theoretical results, unfortunately, it cannot be applied to DNN with the current best hardware configuration. Diakonikolas et al. (2019) uses dimension-free robust mean estimation breakthroughs to design the learning algorithm, while we notice that most robust mean estimation relies on filtering out data by computing the score of projection to the maximum singular vector. For example, in Diakonikolas et al. (2019), it requires performing SVD on $n \times d$ individual gradient matrix, where $n$ is the sample size and $d$ is the number of parameters. This method works well for small datasets and small models since both $n$ and $d$ is small enough for current memory limitation. However, for deep neural networks, this matrix size is far beyond current GPU memory capability. That could be the potential reason why in Diakonikolas et al. (2019), only ridge regression and SVM results for small data are shown (we are not saying that they should provide DNN results). In our experiment, our $n$ is 60000 and $d$ is in the magnitude of millions (network parameters). It is impractical to store 60000 copies of neural networks in a single GPU card. In contrast, in our algorithm, we do not need to store the full gradient matrix. By only considering the loss-layer gradient norm, we can easily extend our algorithm to DNN, and we showed that this simple strategy works well in both theory and challenging empirical tasks.

We notice that there are some linear (Bhatia et al., 2015; 2017) or convex method (Prasad et al., 2018) achieves the better robustness guarantee. However, most of them cannot be directly applied to deep neural networks.

## 4 RELATIONSHIP TO SELF-PACED LEARNING (SPL)

SPL looks very similar to our method at first glance. Instead of keeping data point with small gradient norm, SPL tries to keep data with small loss. The gradient norm and loss function can be tied by the famous Polyak-Łojasiewicz (PL) condition. The PL condition assumes there exists some constant $s > 0$ such that $\frac{1}{2}\|\nabla\phi(\mathbf{x})\|^2 \geq s\left(\phi(\mathbf{x}) - \phi^*\right), \quad \forall \mathbf{x}$ holds. As we can see, when the neural network is highly over-parameterized, the $\phi^*$ can be assumed to be equal across different

samples since neural networks can achieve 0 training loss (Zhang et al., 2016). By sorting the error $\phi(\mathbf{x}_i)$ for every data point, SPL actually is sorting the lower bound of the gradient norm if the PL condition holds. However, the ranking of gradient norm and the ranking loss can be very different since there is no guarantee that the gradient norm is monotonically increasing with the loss value. We provide illustration of why SPL is not robust from geometric perspective in the appendix. Here we show even for simple square loss, the monotonic relationship is easy to break. One easy counter-example is $\phi(x_1, x_2) = 0.5x_1^2 + 50x_2^2$. Take two points (1000, 1) and (495, -49.5), we will find the monotonic relationship does not hold for these two points. Nocedal et al. (2002) showed that the monotonic relationship holds for **square loss** (i.e.$\phi(\mathbf{x}) = \frac{1}{2}(\mathbf{x} - \mathbf{x}^*)^T\mathbf{Q}(\mathbf{x} - \mathbf{x}^*)$ ) if the condition number of $\mathbf{Q}$ is smaller than $3 + 2\sqrt{2}$, which is a quite strong assumption especially when $\mathbf{x}$ is in high-dimension. If we consider the more general type of loss function (i.e. neural network), the assumptions on condition number should only be stronger, thus breaking the monotonic relationship. Thus, although SPL sorts the lower bound of the gradient norm under mild assumptions, our algorithm is significantly different from the proposed SPL and its variations.

Now, we discuss the relationship between SPL and algorithm 3 under supervision corruptions. SPL has the same form as algorithm 3 when we are using mean square error to perform regression tasks since the loss layer gradient norm is equal to loss itself. However, in classification, algorithm 3 is different from the SPL. In order to better understand the algorithm, we further analyze the difference between SPL and our algorithm for cross-entropy loss.

For cross entropy, denote the output logit as $\mathbf{o}$, we have $H(\mathbf{y}_i, \mathbf{f}_i) = -\langle \mathbf{y}_i, \log(\text{softmax}(\mathbf{o}_i))\rangle = -\langle \mathbf{y}_i, \log(\mathbf{f}_i)\rangle$. The gradient norm of cross entropy w.r.t. $\mathbf{o}_i$ is: $\frac{\partial H_i}{\partial \mathbf{o}_i} = \mathbf{y}_i - \text{softmax}(\mathbf{o}_i) = \mathbf{f}_i - \mathbf{y}_i$.

Thus, the gradient of loss layer is the MSE between $\mathbf{y}_i$ and $\mathbf{f}_i$. Next, we investigate when MSE and Cross Entropy gives non-monotonic relationship. For the sake of simplification, we only study the sufficient condition of the non-monotonic relationship, which is showed in lemma 2.

**Lemma 2** *Let $\mathbf{y} \in \mathbb{R}^q$, where $\mathbf{y}_k = 1, \mathbf{y}_i = 0$ for $i \neq k$, and $\alpha, \beta$ are two q dimensional vector in probability simplex. Without loss of generality, suppose $\alpha$ has smaller cross entropy loss $\alpha_k \geq \beta_k$, then the sufficient condition for $\|\alpha - \mathbf{y}\| \geq \|\beta - \mathbf{y}\|$ is $\text{Var}_{i \neq k}(\{\alpha_i\}) - \text{Var}_{i \neq k}(\{\beta_i\}) \geq \frac{q}{(q-1)^2}((\alpha_k - \beta_k)(2 - \alpha_k - \beta_k))$*

As $\alpha_k \geq \beta_k$, the right term is non-negative. In conclusion, when MSE generates a different result from cross-entropy, the variance of the probability of the non-true class of the discarded data point is larger. Suppose we have a ground-truth vector $\mathbf{y} = [0, 1, 0, 0, 0, 0, 0, 0, 0, 0]$, and we have two predictions $\alpha = [0.08, 0.28, 0.08, 0.08, 0.08, 0.08, 0.08, 0.08, 0.08, 0.08]$ and $\beta = [0.1, 0.3, 0.34, 0.05, 0.05, 0.1, 0.03, 0.03, 0, 0]$. The prediction $\alpha$ have a smaller mse loss while prediction $\beta$ have a smaller cross-entropy loss. It is intuitive that $\beta$ is more likely to be noisy data since it has two peak on the prediction (i.e. 0.3, 0.34). However, since cross entropy loss only considers one dimension, it cannot detect such situation. Compared to the cross-entropy, the gradient (mse loss) considers all dimension, and thus will consider the overall prediction distributions.

## 5 COMBINING WITH CO-TEACHING STYLE TRAINING

Motivated by co-teaching (Han et al., 2018), which is one of currently state-of-the-art deep methods for learning under noisy label, we propose *Co-PRL(L)*, which has the same framework of co-teaching but uses the loss-layer gradient to select the data. The full algorithm is shown in algorithm 4 in the appendix. The meaning of all hyper-parameters in algorithm 4 are all the same as in the original Han et al. (2018). Compared with algorithm 3, except sampling data according to the loss layer gradient norm, the *Co-PRL(L)* has two other modules. The first is we gradually increase the amount of the data to be dropped. The second is that two networks will exchange the selected data to update their own parameters.

## 6 EXPERIMENT

In this section, we perform experiments on benchmark regression and classification dataset. The code is available in supplementary materials of submission. We compare *PRL(G)*(Algo. 2), *PRL(L)*

(Algo. 3), and ***Co-PRL(L)*** (Algo. 4) to the following baselines. ***Standard***: standard training without filtering data (mse for regression, cross entropy for classification); ***Normclip***: standard training with norm clipping; ***Huber***: standard training with huber loss (regression only); ***Decouple***: decoupling network, update two networks by using their disagreement (Malach & Shalev-Shwartz, 2017) (classification only); ***Bootstrap***: It uses a weighted combination of predicted and original labels as the correct labels, and then perform back propagation (Reed et al., 2014) (classification only); ***Min-sgd***: choosing the smallest loss sample in minibatch to update model (Shah et al., 2020); ***SPL***: self-paced learning, dropping the data with large losses (same as ***PRL(L)*** in regression setting with MSE loss); ***Ignormclip***: clipping individual gradient then average them to update model (regression only); ***Co-teaching***: collaboratively train a pair of SPL model and exchange selected data to another model(Han et al., 2018) (classification only); It is hard to design experiments for ***agnostic corrupted supervision*** and we tried our best to include different types of supervision noise. The supervision corruption settings are as follows: ***linadv***: the corrupted supervision is generated by random wrong linear relationship of features (regression); ***signflip***: the supervision sign is flipped (regression); ***uninoise***: random sampling from uniform distribution as corrupted supervision (regression); ***mixture***: mixture of above types of corruptions (regression); ***pairflip***: shuffle the coordinates (i.e. eyes to mouth in celebA or cat to dog in CIFAR) (regression and classification); ***symmetric***: randomly assign wrong class label (classification). For classification, we use classification accuracy as the evaluation metric, and R-square is used to evaluate regression experiments. Due to the limit of the space, we only show the average evaluation score on testing data for the last 10 epochs. The whole training curves are attached in the appendix. All experiments are repeated 5 times for regression experiments and 3 times for classification experiments.Main hyperparameters are showed in appendix.

## 6.1 REGRESSION EXPERIMENT

We use CelebA data to perform regression tasks. CelebA dataset has 162770 training images, 19867 validation images, 19962 test images. The target variable is ten-dimensional coordinates of the left eye, right eye, nose, left mouth, and right mouth. Given the human face image, the goal is to predict 10 face landmark coordinates in the image. We tried adding different types of noise on the landmark coordinates. We preprocess the CelebA data as following: we use three-layer CNN to train 162770 training images to predict clean coordinates (we use 19867 validation images to do the early stopping). Then, we use well-trained network to extract the 512-dimensional feature on testing sets. Thus, our final data to perform experiment has feature sets $\mathbf{X} \in \mathbb{R}^{19962 \times 512}$, and the target variable $\mathbf{Y} \in \mathbb{R}^{19962 \times 10}$. We further split the data to the training and testing set, where training sets contain 80% of the data. Then, we manually add ***linadv***, ***signflip***, ***uninoise***, ***pairflip***, ***mixture*** types of supervision noise on the target variable on training data. The corruption rate for all types of corruption is varied from 0.1 to 0.4. We use 3-layer fully connect networks in experiments. The results of averaged last 10 epoch r-square are in table 1.

## 6.2 CLASSIFICATION EXPERIMENT

We perform experiments on CIFAR10, and CIFAR100 to illustrate the effectiveness of our algorithm in classification setting. We use the 9-layer Convolutional Neural Network, which is the same as Han et al. (2018). Since most baselines include batch normalization, it is difficult to get individual gradient efficiently, we will drop the ignormclip and PRL baselines. In the appendix, we attached the results if both co-teaching and Co-PRL(L) drops batch normalization module. We will see that co-teaching cannot maintain robustness while our method still has robustness. The reason is discussed in the appendix. We consider ***pairflip*** and ***symmetric*** supervision corruptions in experiments. Also, to compare with the current state of the art method, for ***symmetric*** noise, we use corruption rate which beyond 0.5. Although our theoretical analysis assumes the noise rate is small than 0.5, when the noise type is not adversary (i.e. symmetric), we empirically show that our method can also deal with such type of noise. Results on CIFAR10, CIFAR100 are in Table 2. As we can see, no matter using one network (PRL vs SPL) or two networks (Co-PRL(L) vs Co-teaching), our method performs significantly better. Since in real-world problems, it is hard to know that the ground-truth corruption rate, we also perform the sensitivity analysis in classification tasks to show the effect of overestimating and underestimating $\epsilon$. The results are in Table 3. More discussion about sensitivity analysis can be found in appendix.

| Corruption | Standard | Normclip | Huber | Min-sgd | Ignormclip | PRL(G) | PRL(L) | Co-PRL(L) |
|---|---|---|---|---|---|---|---|---|
| linadv: 10 | -2.33±0.84 | -2.22±0.74 | 0.868±0.01 | 0.103±0.03 | 0.68±0.07 | **0.876±0.01** | 0.876±0.01 | **0.876±0.01** |
| linadv: 20 | -8.65±2.1 | -8.55±2.2 | 0.817±0.015 | 0.120±0.02 | 0.367±0.28 | **0.871±0.01** | 0.869±0.01 | 0.869±0.01 |
| linadv: 30 | -18.529±4.04 | -19.185±4.31 | 0.592±0.07 | 0.146±0.03 | -0.944±0.51 | **0.865±0.01** | 0.861±0.01 | 0.860±0.01 |
| linadv: 40 | -32.22±6.32 | -32.75±7.07 | -2.529±1.22 | 0.180±0.01 | -1.60 ± 0.80 | **0.857± 0.01** | 0.847±0.02 | 0.847±0.02 |
| signflip: 10 | 0.800±0.02 | 0.798±0.03 | 0.857±0.01 | 0.110±0.04 | 0.846±0.01 | 0.877±0.01 | 0.878±0.01 | **0.879±0.01** |
| signflip: 20 | 0.641±0.05 | 0.638±0.04 | 0.786±0.02 | 0.105±0.07 | 0.82±0.02 | 0.875±0.01 | 0.875±0.01 | **0.877±0.01** |
| signflip: 30 | 0.422±0.04 | 0.421±0.04 | 0.629±0.03 | 0.124±0.05 | 0.795±0.02 | 0.871±0.01 | 0.873±0.01 | **0.875±0.01** |
| signflip: 40 | 0.193±0.043 | 0.190±0.04 | 0.379±0.05 | -0.028±0.25 | 0.759±0.01 | **0.872±0.01** | **0.872±0.01** | 0.871±0.01 |
| uninoise: 10 | 0.845±0.01 | 0.844±0.01 | 0.875±0.01 | 0.103±0.03 | 0.859±0.01 | 0.879±0.01 | **0.881±0.01** | **0.881±0.01** |
| uninoise: 20 | 0.798±0.02 | 0.795±0.02 | 0.865±0.01 | 0.120±0.02 | 0.844±0.01 | 0.878±0.01 | **0.880±0.01** | **0.880±0.01** |
| uninoise: 30 | 0.728±0.02 | 0.725±0.02 | 0.847±0.01 | 0.146±0.03 | 0.831±0.01 | 0.878±0.01 | **0.879±0.01** | **0.879±0.01** |
| uninoise: 40 | 0.656±0.02 | 0.654±0.02 | 0.825±0.01 | 0.180±0.01 | 0.821±0.01 | 0.876± 0.01 | **0.878±0.01** | **0.878±0.01** |
| pairflip: 10 | 0.852±0.02 | 0.851±0.02 | 0.870±0.01 | 0.110±0.04 | 0.867±0.01 | 0.877±0.01 | 0.876±0.01 | **0.878±0.01** |
| pairflip: 20 | 0.784±0.03 | 0.783±0.03 | 0.841±0.02 | 0.120±0.03 | 0.849±0.01 | **0.874±0.01** | 0.873±0.01 | **0.874±0.01** |
| pairflip: 30 | 0.688±0.04 | 0.686±0.04 | 0.770±0.02 | 0.133±0.02 | 0.828±0.01 | 0.870±0.01 | 0.872±0.01 | **0.873±0.01** |
| pairflip: 40 | 0.556±0.06 | 0.553±0.06 | 0.642±0.06 | 0.134±0.03 | 0.810±0.02 | 0.863±0.01 | **0.870±0.01** | **0.870±0.01** |
| mixture: 10 | -0.212±0.6 | -0.010±0.48 | 0.873±0.01 | 0.101±0.03 | 0.861±0.01 | 0.878±0.01 | **0.880±0.01** | **0.880±0.01** |
| mixture: 20 | -0.404±0.68 | -0.463±0.67 | 0.855±0.01 | 0.119±0.03 | 0.855±0.01 | 0.877±0.01 | 0.878±0.01 | **0.879±0.01** |
| mixture: 30 | -0.716±0.57 | -0.824±0.39 | 0.823±0.01 | 0.148±0.02 | 0.847±0.01 | 0.875±0.01 | 0.877±0.01 | **0.878±0.01** |
| mixture: 40 | -3.130±1.51 | -2.69±0.84 | 0.763±0.01 | 0.175±0.02 | 0.835±0.01 | 0.872±0.01 | 0.875 ±0.01 | **0.876±0.01** |

Table 1: CelebA Results. The numbers are r-square on clean testing data, and the standard deviation is from last ten epochs and 5 random seeds.

| Corruption | Standard | Normclip | Bootstrap | Decouple | Min-sgd | SPL | PRL(L) | Co-teaching | Co-PRL(L) |
|---|---|---|---|---|---|---|---|---|---|
| CF10-sym-30 | 63.22 ± 0.18 | 62.41 ± 0.06 | 63.67 ± 0.24 | 70.73 ± 0.51 | 13.31 ± 2.24 | 77.77 ± 0.34 | 79.40 ± 0.19 | 79.90 ± 0.13 | **80.05 ± 0.12** |
| CF10-sym-50 | 44.63 ± 0.18 | 43.99 ± 0.28 | 46.13 ± 0.18 | 57.48 ± 1.98 | 13.33 ± 2.85 | 72.22 ± 0.15 | 74.17 ± 0.15 | 74.25 ± 0.41 | **75.43 ± 0.09** |
| CF10-sym-70 | 24.12 ± 0.09 | 24.17 ± 0.37 | 25.13 ± 0.39 | 40.11 ± 4.62 | 9.08 ± 0.94 | 56.19 ± 0.33 | 58.36 ± 0.62 | 58.41 ± 0.33 | **60.26 ± 0.42** |
| CF10-pf-25 | 68.34 ± 0.30 | 67.92 ± 0.43 | 68.71 ± 0.32 | 75.59 ± 0.35 | 10.45 ± 0.60 | 75.79 ± 0.44 | 80.54 ± 0.07 | 80.18 ± 0.21 | **81.51 ± 0.13** |
| CF10-pf-35 | 58.68 ± 0.28 | 58.27 ± 0.18 | 58.19 ± 0.12 | 66.38 ± 0.44 | 12.29 ± 1.92 | 70.40 ± 0.27 | 77.61 ± 0.35 | 77.97 ± 0.03 | **79.01 ± 0.14** |
| CF10-pf-45 | 48.05 ± 0.25 | 48.03 ± 0.54 | 47.84 ± 0.32 | 51.54 ± 0.81 | 10.94 ± 1.28 | 58.95 ± 0.59 | 71.42 ± 0.24 | 72.43 ± 0.31 | **73.78 ± 0.17** |
| CF100-sym-30 | 32.83 ± 0.39 | 32.10 ± 0.64 | 34.47 ± 0.22 | 32.95 ± 0.44 | 2.94 ± 0.61 | 44.37 ± 0.44 | 46.40 ± 0.18 | 45.02 ± 0.29 | **47.51 ± 0.47** |
| CF100-sym-50 | 20.47 ± 0.44 | 19.73 ± 0.29 | 21.59 ± 0.44 | 21.02 ± 0.36 | 2.35 ± 0.45 | 37.89 ± 0.16 | 38.38 ± 0.65 | 38.79 ± 0.33 | **40.64 ± 0.11** |
| CF100-sym-70 | 9.93 ± 0.07 | 9.93 ± 0.23 | 10.59 ± 0.17 | 12.55 ± 0.46 | 2.32 ± 0.24 | 24.10 ± 0.44 | 25.38 ± 0.56 | 24.94 ± 0.53 | **27.27 ± 0.01** |
| CF100-pf-25 | 40.37 ± 0.55 | 39.34 ± 0.35 | 40.22 ± 0.37 | 39.43 ± 0.27 | 2.62 ± 0.26 | 40.48 ± 0.72 | 47.57 ± 0.37 | 42.97 ± 0.10 | **48.06 ± 0.26** |
| CF100-pf-35 | 34.07 ± 0.19 | 32.88 ± 0.10 | 34.53 ± 0.23 | 33.14 ± 0.07 | 2.30 ± 0.07 | 34.17 ± 0.46 | 43.32 ± 0.16 | 36.69 ± 0.23 | **44.08 ± 0.33** |
| CF100-pf-45 | 27.66 ± 0.50 | 27.35 ± 0.61 | 27.56 ± 0.23 | 26.83 ± 0.41 | 2.55 ± 0.52 | 27.55 ± 0.66 | 33.31 ± 0.10 | 29.71 ± 0.20 | **34.43 ± 0.05** |

Table 2: Classification Results on CIFAR10 and CIFAR100 for *symmetric* and *pairflip* label corruption. The numbers are classification accuracy on clean testing data, and the standard deviation is from last ten epochs and 3 random seeds.

| Data | $\epsilon - 0.1$ | $\epsilon - 0.05$ | $\epsilon$ | $\epsilon + 0.05$ | $\epsilon + 0.1$ |
|---|---|---|---|---|---|
| CF10-Pair-45% | 65.07±0.83 | 70.07±0.67 | 73.78±0.17 | 77.56±0.55 | 79.36±0.43 |
| CF10-Sym-50% | 69.21±0.35 | 72.53±0.45 | 75.43 ± 0.09 | 77.65±0.27 | 78.10±0.31 |
| CF10-Sym-70% | 53.88±0.64 | 58.49±0.97 | 60.26 ± 0.42 | 60.89±0.43 | 54.91±0.68 |
| CF100-Pair-45% | 32.60±0.45 | 34.17±0.40 | 34.43 ± 0.05 | 36.87±0.41 | 38.34±0.78 |
| CF100-Sym-50% | 37.74±0.41 | 39.72±0.36 | 40.64 ± 0.11 | 43.02±0.36 | 43.92±0.61 |
| CF100-Sym-70% | 24.40±0.47 | 25.50±0.45 | 27.27 ± 0.10 | 27.80±0.50 | 28.20±0.97 |

Table 3: sensitivity analysis for estimated $\epsilon$

## 7 CONCLUSION

In this paper, we proposed efficient algorithm to defense against agnostic supervision corruptions. Both theoratical and empirical analysis showed the effectiveness of our algorithm. There are two remaining questions in this paper which deserves study in future. The first one is whether we can further improve $\mathcal{O}(\epsilon)$ error bound or show that $\mathcal{O}(\epsilon)$ is tight. The second one is to utilize more properties of neural networks, such as the sparse gradient, to see whether it is possible to get better algorithms.

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

## A APPENDIX

### A.1 CO-IGFILTER ALGORITHM

See algorithm 4.

---

**Algorithm 4:** Co-PRL(L)

---

**input:** initialize $w_f$ and $w_g$, learning rate $\eta$, fixed $\tau$, epoch $T_k$ and $T_{max}$, iterations $N_{max}$
**return** *model parameter $w_f$ and $w_g$;*
**for** $T = 1, 2, ..., T_{max}$ **do**
    **for** $N = 1, ..., N_{max}$ **do**
        random sample a minibatch $\mathbf{M}$ from $\mathbf{D}_x, \mathbf{D}_y^\epsilon$      (noisy dataset)
        get the predicted label $\hat{\mathbf{Y}}_f$ and $\hat{\mathbf{Y}}_g$ from $\mathbf{M}$ by $w_f$. $w_g$
        calculate the individual loss $l_f = \mathcal{L}(\mathbf{Y}, \hat{\mathbf{Y}}_f), l_g = \mathcal{L}(\mathbf{Y}, \hat{\mathbf{Y}}_g)$
        calculate the gradient norm of loss layer $score_f = \|\frac{\partial l_f}{\partial \hat{\mathbf{y}}_f}\|, score_g = \|\frac{\partial l_g}{\partial \hat{\mathbf{y}}_g}\|$.
        sample $R(T)\%$ small-loss-layer-gradient-norm instances by $score_f$ and $score_g$ to get $\mathbf{N}_f, \mathbf{N}_g$
        update $w_f = w_f - \eta\nabla_{w_f}\mathcal{L}(\mathbf{N}_f, w_f), w_g = w_g - \eta\nabla_{w_g}\mathcal{L}(\mathbf{N}_g, w_g)$     (selected dataset)
        update model $\mathbf{x}_{t+1} = \mathbf{x}_t - \gamma_t\hat{\mu}$
    **end**
    **Update** $R(T) = 1 - \min\left\{\frac{T}{T_k}\tau, \tau\right\}$
**end**

---

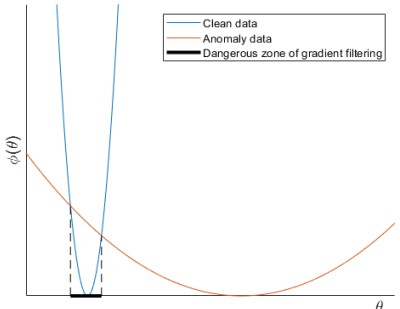 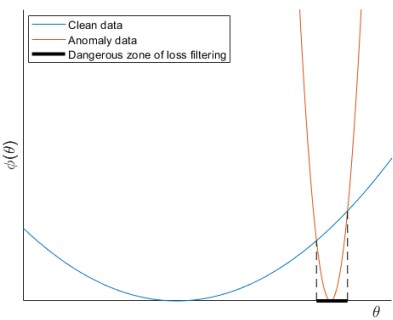

(a) When gradient filtering method failed to pick out right corrupted data, the remaining corrupted data is relatively smooth, thus has limited impact on overall loss surface.

(b) When loss filtering method failed to pick out right corrupted data, the remaining corrupted data could be extremely sharp, thus has large impact on overall loss surface.

Figure 1: Geometric illustration of the difference between loss filtering and gradient norm filtering.

## A.2 FURTHER ILLUSTRATION OF THE DIFFERENCE BETWEEN SPL AND PRL(G)

In this section, we will further illustrate the difference between SPL and PRL(G). In order to have a more intuitive understanding of our algorithm, we could look at the Figure 1a and 1b. Since we are in the agnostic label corruption setting, it is difficult to filtering out the correct corrupted data. We showed two situations when loss filtering failed and gradient filtering failed. As we could see that when loss filtering method failed, the remaining corrupted data could have large impact on the overall loss surface while when gradient filtering method failed, the remaining corrupted data only have limited impact on the overall loss surface, thus gaining robustness.

## A.3 NETWORKS AND HYPERPARAMETERS

The hyperparameters are in Table 4. For Classification, we use the same hyperparameters in Han et al. (2018). For CelebA, we use 3-layer fully connected network with 256 hidden nodes in hidden layer and leakly-relu as activation function. We also attached our code in supplementary materials.

| Data\HyperParameter | BatchSize | Learning Rate | Optimizer | Momentum |
|---|---|---|---|---|
| CF-10 | 128 | 0.001 | Adam | 0.9 |
| CF-100 | 128 | 0.001 | Adam | 0.9 |
| CelebA | 512 | 0.0003 | Adam | 0.9 |

Table 4: Main Hyperparmeters

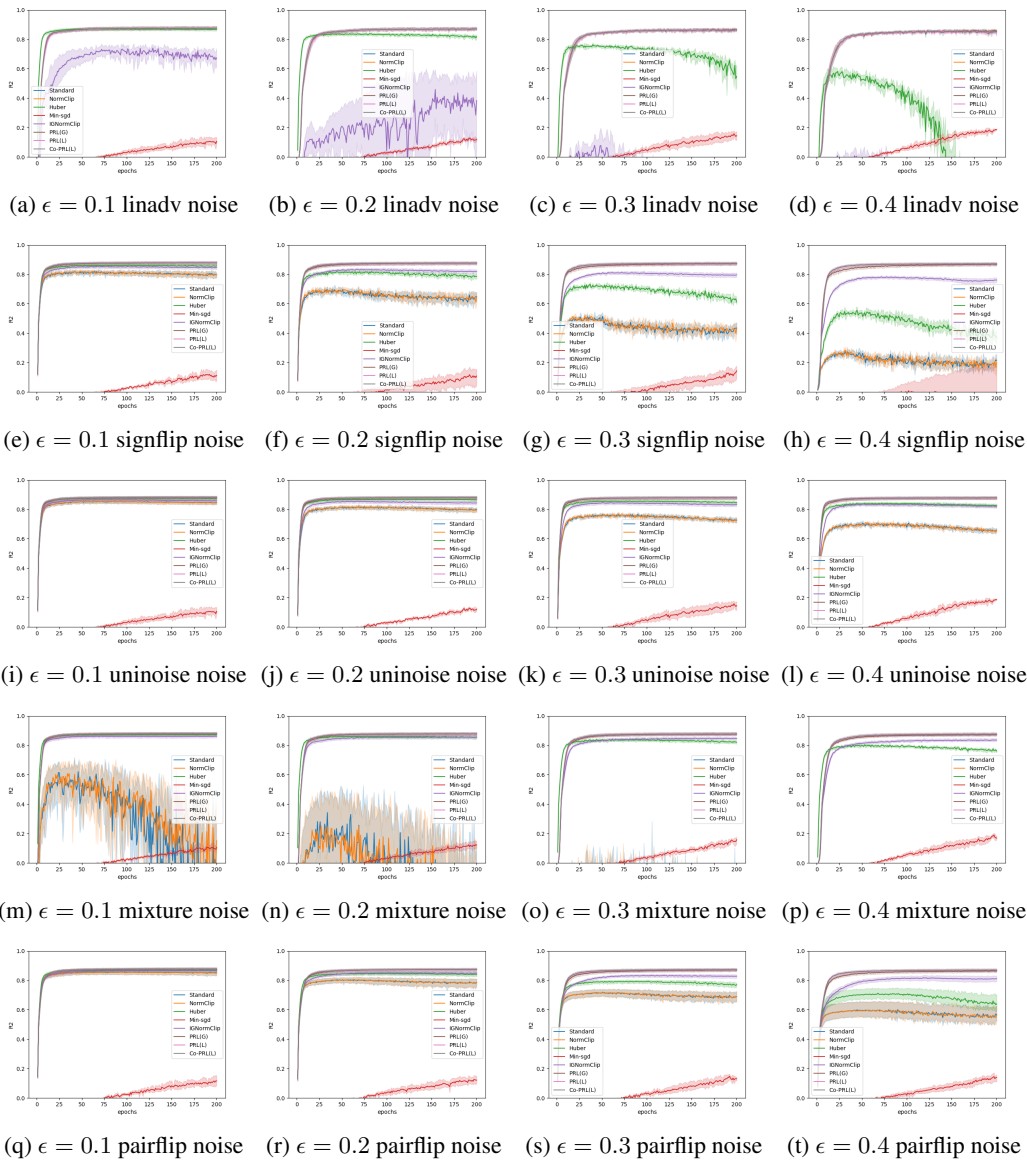

Figure 2: CelebA Testing Curve During Training. X axis represents the epoch number, Y axis represents the testing r-square. In some experiment, there is no cure for Standard and NormClip since they gave negative r-square, which will effect the plotting scale. The shadow represents the confidence interval, which is calculated across 5 random seed. As we see, PRL(G), PRL(L), and Co-PRL(L) are robust against different types of corruptions.

## A.4 REGRESSION R2 ON TESTING DATA CURVE

The curve for CelebA data is showed in Figure 2.

## A.5 CLASSIFICATION CURVE

The classification curve is in Figure 3

## A.6 SENSITIVITY ANALYSIS

Since in real-world problems, it is hard to know that the ground-truth corruption rate, we perform the sensitivity analysis in classification tasks to show the effect of $\epsilon$. The results are in Table 5. As we could see, the performance is stable if we overestimate the corruption rate, this is because only

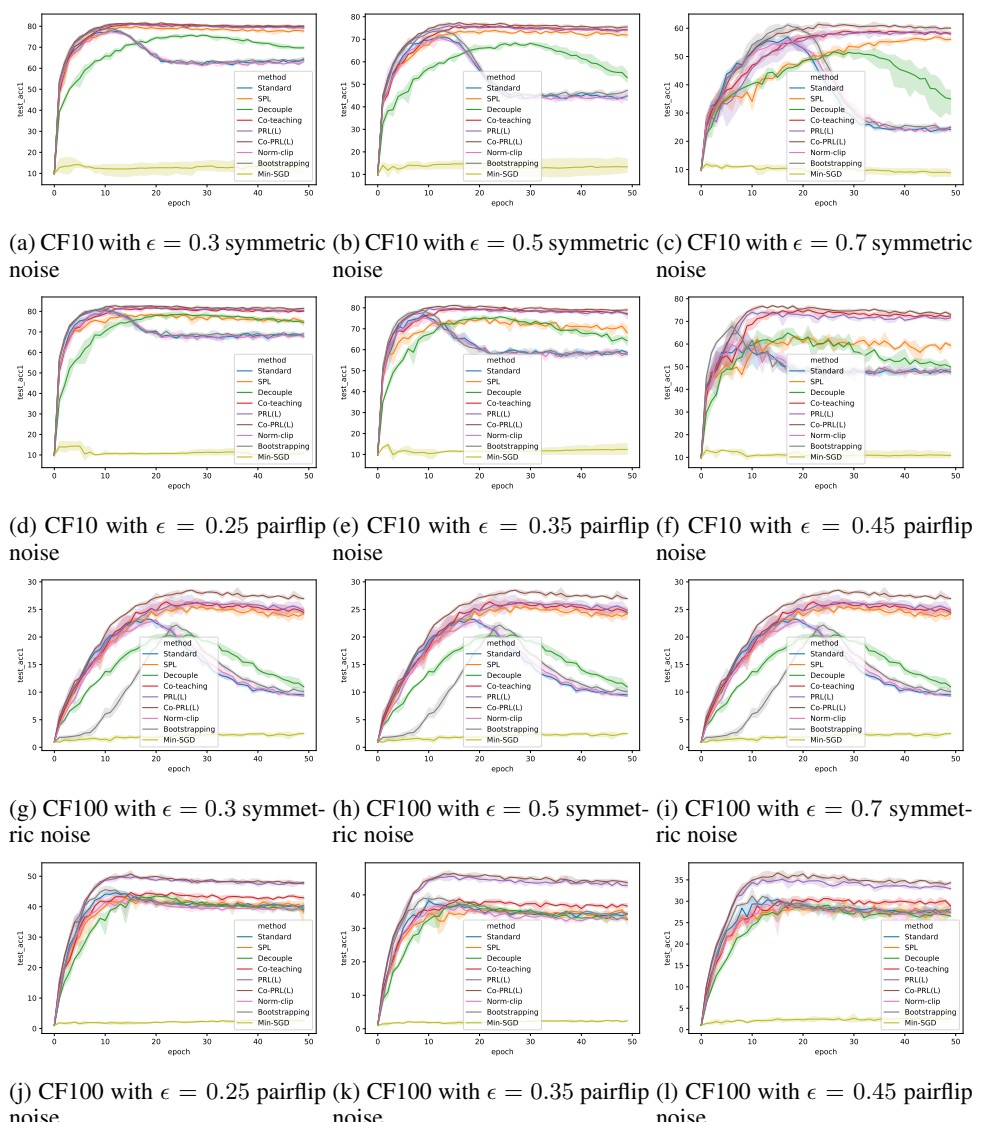

(a) CF10 with $\epsilon = 0.3$ symmetric noise (b) CF10 with $\epsilon = 0.5$ symmetric noise (c) CF10 with $\epsilon = 0.7$ symmetric noise

(d) CF10 with $\epsilon = 0.25$ pairflip noise (e) CF10 with $\epsilon = 0.35$ pairflip noise (f) CF10 with $\epsilon = 0.45$ pairflip noise

(g) CF100 with $\epsilon = 0.3$ symmetric noise (h) CF100 with $\epsilon = 0.5$ symmetric noise (i) CF100 with $\epsilon = 0.7$ symmetric noise

(j) CF100 with $\epsilon = 0.25$ pairflip noise (k) CF100 with $\epsilon = 0.35$ pairflip noise (l) CF100 with $\epsilon = 0.45$ pairflip noise

Figure 3: CIFAR10 and CIFAR100 Testing Curve During Training. X axis represents the epoch number, Y axis represents the testing accuracy. The shadow represents the confidence interval, which is calculated across 3 random seed. As we see, IGFilter(L), and Co-IGFilter(L) are robust against different types of corruptions.

when we overestimate the $\epsilon$, we could guarantee that the gradient norm of the remaining set is small. However, when we underestimate the corruption rate, in the worst case, there is no guarantee that the gradient norm of the remaining set is small. By using the empirical mean, even one large bad individual gradient would ruin the gradient estimation, and according to the convergence analysis of biased gradient descent, the final solution could be very bad in terms of clean data. That explains why to underestimate the corruption rate gives bad results. Also, from Table 5, we could see that using the ground truth corruption rate will lead to small uncertainty.

## A.7 EMPIRICAL RESULTS ON RUNNING TIME

As we claimed in paper, the algorithm 2 is not efficient. In here we attached the execution time for one epoch for three different methods: **Standard**, **PRL(G)**, **PRL(L)**. For fair comparison, we replace all batch normalization module to group normalization for this comparison, since it is hard

| Data | $\epsilon - 0.1$ | $\epsilon - 0.05$ | $\epsilon$ | $\epsilon + 0.05$ | $\epsilon + 0.1$ |
|---|---|---|---|---|---|
| CF10-Pair-45% | 65.07±0.83 | 70.07±0.67 | 73.78±0.17 | 77.56±0.55 | 79.36±0.43 |
| CF10-Sym-50% | 69.21±0.35 | 72.53±0.45 | 75.43 ± 0.09 | 77.65±0.27 | 78.10±0.31 |
| CF10-Sym-70% | 53.88±0.64 | 58.49±0.97 | 60.26 ± 0.42 | 60.89±0.43 | 54.91±0.68 |
| CF100-Pair-45% | 32.60±0.45 | 34.17±0.40 | 34.43 ± 0.05 | 36.87±0.41 | 38.34±0.78 |
| CF100-Sym-50% | 37.74±0.41 | 39.72±0.36 | 40.64 ± 0.11 | 43.02±0.36 | 43.92±0.61 |
| CF100-Sym-70% | 24.40±0.47 | 25.50±0.45 | 27.27 ± 0.10 | 27.80±0.50 | 28.20±0.97 |

Table 5: sensitivity analysis for estimated $\epsilon$

| Method | Standard (Lower Bound) | PRL(G) | PRL(L) |
|---|---|---|---|
| CF10-Pair-45% | 37.03s | 145.55s | 54.80s |

Table 6: Execution Time of Single Epoch in CIFAR-10 Data

to calculate individual gradient when using batch normalization. For PRL(G), we use opacus libarary (https://opacus.ai/) to calculate the individual gradient.

The results are showed in Table 6

### A.8    PROOF OF CONVERGENCE OF BIASED SGD

We gave the proof of the theorem of how biased gradient affect the final convergence of SGD. We introduce several assumptions and definition first:

**Assumption 2 (L-smoothness)** *The function $\phi$: $\mathbb{R}^d \to \mathbb{R}$ is differentiable and there exists a constant $L > 0$ such that for all $\theta_1, \theta_2 \in \mathbb{R}^d$, we have $\phi(\theta_2) \leq \phi(\theta_1) + \langle \nabla\phi(\theta_1), \theta_2 - \theta_1 \rangle + \frac{L}{2}\|\theta_2 - \theta_1\|^2$*

**Definition 1 (Biased gradient oracle)** *A map $\mathbf{g} : \mathbb{R}^d \times \mathcal{D} \to \mathbb{R}^d$, such that $\mathbf{g}(\theta, \xi) = \nabla\phi(\theta) + \mathbf{b}(\theta, \xi) + \mathbf{n}(\theta, \xi)$ for a bias $\mathbf{b} : \mathbb{R}^d \to \mathbb{R}^d$ and zero-mean noise $\mathbf{n} : \mathbb{R}^d \times \mathcal{D} \to \mathbb{R}^d$, that is $\mathbb{E}_\xi \mathbf{n}(\theta, \xi) = 0$.*

Compared to standard stochastic gradient oracle, the above definition introduces the bias term $\mathbf{b}$. In noisy-label settings, the $\mathbf{b}$ is generated by the data with corrupted labels.

**Assumption 3 ($\sigma$-Bounded noise)** *There exists constants $\sigma > 0$, such that $\mathbb{E}_\xi \|\mathbf{n}(\theta, \xi)\|^2 \leq \sigma, \quad \forall \theta \in \mathbb{R}^d$*

**Assumption 4 ($\zeta$-Bounded bias)** *There exists constants $\zeta > 0$, such that for any $\xi$, we have $\|\mathbf{b}(\theta, \xi)\|^2 \leq \zeta^2, \quad \forall \theta \in \mathbb{R}^d$*

For simplicity, assume the learning rate is constant $\gamma$, then in every iteration, the biased SGD performs update $\theta_{t+1} \leftarrow \theta_t - \gamma_t \mathbf{g}(\theta_t, \xi)$. Then the following theorem showed the gradient norm convergence with biased SGD.

**Theorem 4 (Convergence of Biased SGD(formal))** *Under assumptions 2, 3, 4, define $F = \phi(\theta_0) - \phi^*$ and step size $\gamma = \min\left\{\frac{1}{L}, (\sqrt{\frac{LF}{\sigma T}})\right\}$, denote the desired accuracy as $k$, then*

$$T = \mathcal{O}\left(\frac{1}{k} + \frac{\sigma^2}{k^2}\right)$$

*iterations are sufficient to obtain $\min_{t \in [T]} \mathbb{E}\|\nabla\phi(\theta_t)\|^2 = \mathcal{O}(k + \zeta^2)$.*

**Remark 2** *Let $k = \zeta^2$, $T = \mathcal{O}\left(\frac{1}{\zeta^2} + \frac{\sigma^2}{\zeta^4}\right)$ iterations is sufficient to get $\min_{t \in [T]} \mathbb{E}\|\nabla\phi(\theta_t)\|^2 = \mathcal{O}(\zeta^2)$, and performing more iterations does not improve the accuracy in terms of convergence.*

Since this is a standard results, similar results are showed in Bernstein et al. (2018); Devolder et al. (2014); Hu et al. (2020); Ajalloeian & Stich (2020). we provide the proof here.
Proof: by L-smooth, we have:

$$\phi(\theta_2) \leq \phi(\theta_1) + \langle \nabla\phi(\theta_1), \theta_2 - \theta_1 \rangle + \frac{L}{2}\|\theta_2 - \theta_1\|^2$$

by using $\gamma \leq \frac{1}{L}$, we have

$$\mathbb{E}\phi\left(\theta_{\mathbf{1}t+1}\right) \leq \phi\left(\theta_{\mathbf{1}t}\right) - \gamma\left\langle\nabla\phi\left(\theta_{\mathbf{1}t}\right), \mathbb{E}\mathbf{g}_t\right\rangle + \frac{\gamma^2 L}{2}\left(\mathbb{E}\left\|\mathbf{g}_t - \mathbb{E}\mathbf{g}_t\right\|^2 + \mathbb{E}\left\|\mathbb{E}\mathbf{g}_t\right\|^2\right)$$

$$= \phi\left(\theta_{\mathbf{1}t}\right) - \gamma\left\langle\nabla\phi\left(\theta_{\mathbf{1}t}\right), \nabla\phi\left(\theta_{\mathbf{1}t}\right) + \mathbf{b}_t\right\rangle + \frac{\gamma^2 L}{2}\left(\mathbb{E}\left\|\mathbf{n}_t\right\|^2 + \mathbb{E}\left\|\nabla\phi\left(\theta_{\mathbf{1}t}\right) + \mathbf{b}_t\right\|^2\right)$$

$$\leq \phi\left(\theta_{\mathbf{1}t}\right) + \frac{\gamma}{2}\left(-2\left\langle\nabla\phi\left(\theta_{\mathbf{1}t}\right), \nabla\phi\left(\theta_{\mathbf{1}t}\right) + \mathbf{b}_t\right\rangle + \left\|\nabla\phi\left(\theta_{\mathbf{1}t}\right) + \mathbf{b}_t\right\|^2\right) + \frac{\gamma^2 L}{2}\mathbb{E}\left\|\mathbf{n}_t\right\|^2$$

$$= \phi\left(\theta_{\mathbf{1}t}\right) + \frac{\gamma}{2}\left(-\left\|\nabla\phi\left(\theta_{\mathbf{1}t}\right)\right\|^2 + \left\|\mathbf{b}_t\right\|^2\right) + \frac{\gamma^2 L}{2}\mathbb{E}\left\|\mathbf{n}_t\right\|^2$$

Since we have $\|\mathbf{b}_t\|^2 \leq \zeta^2$, $\|\mathbf{n}_t\|^2 \leq \sigma^2$, by plug in the learning rate constraint, we have

$$\mathbb{E}\phi\left(\theta_{\mathbf{1}t+1}\right) \leq \phi\left(\theta_{\mathbf{1}t}\right) - \frac{\gamma}{2}\left\|\nabla\phi\left(\theta_{\mathbf{1}t}\right)\right\|^2 + \frac{\gamma}{2}\zeta^2 + \frac{\gamma^2 L}{2}\sigma^2$$

$$\mathbb{E}\phi\left(\theta_{\mathbf{1}t+1}\right) - \phi\left(\theta_{\mathbf{1}t}\right) \leq -\frac{\gamma}{2}\left\|\nabla\phi\left(\theta_{\mathbf{1}t}\right)\right\|^2 + \frac{\gamma}{2}\zeta^2 + \frac{\gamma^2 L}{2}\sigma^2$$

Then, removing the gradient norm to left hand side, and sum it across different iterations, we could get

$$\frac{1}{2T}\sum_{t=0}^{T-1}\mathbb{E}\|\phi\left(\theta_{\mathbf{1}t}\right)\| \leq \frac{F}{T\gamma} + \frac{\zeta^2}{2} + \frac{\gamma L\sigma^2}{2}$$

Take the minimum respect to t and substitute the learning rate condition will directly get the results.

### A.9 PROOF OF THEOREM 2

Denote $\tilde{\mathbf{G}}$ to be the set of corrupted minibatch, $\mathbf{G}$ to be the set of original clean minibatch and we have $|\mathbf{G}| = |\tilde{\mathbf{G}}| = m$. Let $\mathbf{N}$ to be the set of remaining data and according to our algorithm, the remaining data has the size $|\mathbf{N}| = n = (1-\epsilon)m$. Define $\mathbf{A}$ to be the set of individual clean gradient, which is not discarded by algorithm 1. $\mathbf{B}$ to be the set of individual corrupted gradient, which is not discarded. According to our definition, we have $\mathbf{N} = \mathbf{A} \cup \mathbf{B}$. $\mathbf{AD}$ to be the set of individual good gradient, which is discarded, $\mathbf{AR}$ to be the set of individual good gradient, which is replaced by corrupted data. We have $\mathbf{G} = \mathbf{A} \cup \mathbf{AD} \cup \mathbf{AR}$. $\mathbf{BD}$ is the set of individual corrupted gradient, which is discarded by our algorithm. Denote the good gradient to be $\mathbf{g}_i = \alpha_i \mathbf{W}_i$, and the bad gradient to be $\tilde{\mathbf{g}}_i$, according to our assumption, we have $\|\tilde{\mathbf{g}}_i\| \leq L$.

Now, we have the l2 norm error:

$$\|\mu(\mathbf{G}) - \mu(\mathbf{N})\| = \|\frac{1}{m}\sum_{i\in\mathbf{G}}^{m}\mathbf{g}_i - \left(\frac{1}{n}\sum_{i\in\mathbf{A}}\mathbf{g}_i + \frac{1}{n}\sum_{i\in\mathbf{B}}\tilde{\mathbf{g}}_i\right)\|$$

$$= \|\frac{1}{n}\sum_{i=1}^{m}\frac{n}{m}\mathbf{g}_i - \left(\frac{1}{n}\sum_{i\in\mathbf{A}}\mathbf{g}_i + \frac{1}{n}\sum_{i\in\mathbf{B}}\tilde{\mathbf{g}}_i\right)\|$$

$$= \|\frac{1}{n}\sum_{i\in\mathbf{A}}\frac{n}{m}\mathbf{g}_i + \frac{1}{n}\sum_{i\in\mathbf{AD}}\frac{n}{m}\mathbf{g}_i + \frac{1}{n}\sum_{i\in\mathbf{AR}}\frac{n}{m}\mathbf{g}_i - \left(\frac{1}{n}\sum_{i\in\mathbf{A}}\mathbf{g}_i + \frac{1}{n}\sum_{i\in\mathbf{B}}\tilde{\mathbf{g}}_i\right)\|$$

$$= \|\frac{1}{n}\sum_{i\in\mathbf{A}}(\frac{n-m}{m})\mathbf{g}_i + \frac{1}{n}\sum_{i\in\mathbf{AD}}\frac{n}{m}\mathbf{g}_i + \frac{1}{n}\sum_{i\in\mathbf{AR}}\frac{n}{m}\mathbf{g}_i - \frac{1}{n}\sum_{i\in\mathbf{B}}\tilde{\mathbf{g}}_i\|$$

$$\leq \|\frac{1}{n}\sum_{i\in\mathbf{A}}(\frac{n-m}{m})\mathbf{g}_i + \frac{1}{n}\sum_{i\in\mathbf{AD}}\frac{n}{m}\mathbf{g}_i + \frac{1}{n}\sum_{i\in\mathbf{AR}}\frac{n}{m}\mathbf{g}_i\| + \|\frac{1}{n}\sum_{i\in\mathbf{B}}\tilde{\mathbf{g}}_i\|$$

$$\leq \|\sum_{\mathbf{A}}\frac{m-n}{nm}\mathbf{g}_i + \sum_{\mathbf{AD}}\frac{1}{m}\mathbf{g}_i + \sum_{\mathbf{AR}}\frac{1}{m}\mathbf{g}_i\| + \|\sum_{\mathbf{B}}\frac{1}{n}\tilde{\mathbf{g}}_i\|$$

$$\leq \sum_{\mathbf{A}}\|\frac{m-n}{nm}\mathbf{g}_i\| + \sum_{\mathbf{AD}}\|\frac{1}{m}\mathbf{g}_i\| + \sum_{\mathbf{AR}}\|\frac{1}{m}\mathbf{g}_i\| + \sum_{\mathbf{B}}\frac{1}{n}\|\tilde{\mathbf{g}}_i\|$$

By using the filtering algorithm, we could guarantee that $\|\tilde{\mathbf{g}}_i\| \leq L$. Let $|\mathbf{A}| = x$, we have $|\mathbf{B}| = n - x = (1 - \epsilon)m - x$, $|\mathbf{AR}| = m - n = \epsilon m$, $|\mathbf{AD}| = m - |\mathbf{A}| - |\mathbf{AR}| = m - x - (m - n) = n - x = (1 - \epsilon)m - x$. Thus, we have:

$$
\begin{aligned}
\|\mu(\mathbf{G}) - \mu(\mathbf{N})\| &\leq x\frac{m - n}{nm}L + (n - x)\frac{1}{m}L + (m - n)\frac{1}{m}L + (n - x)\frac{1}{n}L \\
&\leq x(\frac{m - n}{nm} - \frac{1}{m})L + n\frac{1}{m}L + (m - n)\frac{1}{m}L + (n - x)\frac{1}{n}L \\
&= \frac{1}{m}(\frac{2\epsilon - 1}{1 - \epsilon})xL + L + L - \frac{1}{n}xL \\
&= xL(\frac{2\epsilon - 2}{n}) + 2L
\end{aligned}
$$

To minimize the upper bound, we need $x$ to be as small as possible since $2\epsilon - 2 < 1$. According to our problem setting, we have $x = n - m\epsilon \leq (1 - 2\epsilon)m$, substitute back we have:

$$
\begin{aligned}
\|\mu(\mathbf{G}) - \mu(\mathbf{N})\| &\leq (1 - 2\epsilon)Lm(\frac{2\epsilon - 2}{n}) + 2L \\
&= \frac{1 - 2\epsilon}{1 - \epsilon}2L + 2L \\
&= 4L - \frac{\epsilon}{1 - \epsilon}2L
\end{aligned}
$$

Since $\epsilon < 0.5$, we use tylor expansion on $\frac{\epsilon}{1 - \epsilon}$, by ignoring the high-order terms, we have

$$
\|\mu(\mathbf{G}) - \mu(\mathbf{N})\| = \mathcal{O}(\epsilon L)
$$

Note, if the Lipschitz continuous assumption does not hold, then L should be dimension dependent.

## A.10 PROOF OF RANDOMIZED FILTERING ALGORITHM

**Lemma 3 (Gradient Estimation Error for Randomized Filtering)** *Given a corrupted matrix $\tilde{\mathbf{G}} \in \mathbb{R}^{m \times d}$ generated in problem 2. Let $\mathbf{G} \in \mathbb{R}^{m \times d}$ be the original clean gradient matrix. Suppose we are arbitrary select $n = (1 - \epsilon)m$ rows from $\tilde{\mathbf{G}}$ to get remaining set $\mathbf{N} \in \mathbb{R}^{n \times d}$. Let $\mu$ to be the empirical mean function, assume the clean gradient before loss layer has bounded operator norm: $\|\mathbf{W}\|_{op} \leq C$, the maximum clean gradient in loss layer $\max_i \|\alpha_i\| = k$, the maximum corrupted gradient in loss layer $\max_i \|\delta_i\| = v$, assume $\epsilon < 0.5$, then we have:*

$$
\|\mu(\mathbf{G}) - \mu(\mathbf{N})\| \leq Ck\frac{3\epsilon - 4\epsilon^2}{1 - \epsilon} + Cv\frac{\epsilon}{1 - \epsilon}
$$

### A.10.1 PROOF OF LEMMA 3

Denote $\tilde{\mathbf{G}}$ to be the set of corrupted minibatch, $\mathbf{G}$ to be the set of original clean minibatch and we have $|\mathbf{G}| = |\tilde{\mathbf{G}}| = m$. Let $\mathbf{N}$ to be the set of remaining data and according to our algorithm, the remaining data has the size $|\mathbf{N}| = n = (1 - \epsilon)m$. Define $\mathbf{A}$ to be the set of individual clean gradient, which is not discarded by algorithm 3. $\mathbf{B}$ to be the set of individual corrupted gradient, which is not discarded. According to our definition, we have $\mathbf{N} = \mathbf{A} \cup \mathbf{B}$. $\mathbf{AD}$ to be the set of individual good gradient, which is discarded, $\mathbf{AR}$ to be the set of individual good gradient, which is replaced by corrupted data. We have $\mathbf{G} = \mathbf{A} \cup \mathbf{AD} \cup \mathbf{AR}$. $\mathbf{BD}$ is the set of individual corrupted gradient, which is discarded by our algorithm. Denote the good gradient to be $\mathbf{g}_i = \alpha_i \mathbf{W}_i$, and the bad gradient to be $\tilde{\mathbf{g}}_i = \delta_i \mathbf{W}_i$, according to our assumption, we have $\|\mathbf{W}_i\|_{op} \leq C$.

Now, we have the l2 norm error:

$$
\|\mu(\mathbf{G}) - \mu(\mathbf{N})\| = \|\frac{1}{m}\sum_{i\in\mathbf{G}}^{m}\mathbf{g}_i - \left(\frac{1}{n}\sum_{i\in\mathbf{A}}\mathbf{g}_i + \frac{1}{n}\sum_{i\in\mathbf{B}}\tilde{\mathbf{g}}_i\right)\|
$$

$$
= \|\frac{1}{n}\sum_{i=1}^{m}\frac{n}{m}\mathbf{g}_i - \left(\frac{1}{n}\sum_{i\in\mathbf{A}}\mathbf{g}_i + \frac{1}{n}\sum_{i\in\mathbf{B}}\tilde{\mathbf{g}}_i\right)\|
$$

$$
= \|\frac{1}{n}\sum_{i\in\mathbf{A}}\frac{n}{m}\mathbf{g}_i + \frac{1}{n}\sum_{i\in\mathbf{AD}}\frac{n}{m}\mathbf{g}_i + \frac{1}{n}\sum_{i\in\mathbf{AR}}\frac{n}{m}\mathbf{g}_i - \left(\frac{1}{n}\sum_{i\in\mathbf{A}}\mathbf{g}_i + \frac{1}{n}\sum_{i\in\mathbf{B}}\tilde{\mathbf{g}}_i\right)\|
$$

$$
= \|\frac{1}{n}\sum_{i\in\mathbf{A}}(\frac{n-m}{m})\mathbf{g}_i + \frac{1}{n}\sum_{i\in\mathbf{AD}}\frac{n}{m}\mathbf{g}_i + \frac{1}{n}\sum_{i\in\mathbf{AR}}\frac{n}{m}\mathbf{g}_i - \frac{1}{n}\sum_{i\in\mathbf{B}}\tilde{\mathbf{g}}_i\|
$$

$$
\leq \|\frac{1}{n}\sum_{i\in\mathbf{A}}(\frac{n-m}{m})\mathbf{g}_i + \frac{1}{n}\sum_{i\in\mathbf{AD}}\frac{n}{m}\mathbf{g}_i + \frac{1}{n}\sum_{i\in\mathbf{AR}}\frac{n}{m}\mathbf{g}_i\| + \|\frac{1}{n}\sum_{i\in\mathbf{B}}\tilde{\mathbf{g}}_i\| \qquad (1)
$$

Let $|\mathbf{A}| = x$, we have $|\mathbf{B}| = n-x = (1-\epsilon)m-x$, $|\mathbf{AR}| = m-n = \epsilon m$, $|\mathbf{AD}| = m-|\mathbf{A}|-|\mathbf{AR}| = m-x-(m-n) = n-x = (1-\epsilon)m-x$. Thus, we have:

$$
\|\mu(\mathbf{G}) - \mu(\mathbf{N})\| \leq \|\sum_{\mathbf{A}}\frac{m-n}{nm}\mathbf{g}_i + \sum_{\mathbf{AD}}\frac{1}{m}\mathbf{g}_i + \sum_{\mathbf{AR}}\frac{1}{m}\mathbf{g}_i\| + \sum_{\mathbf{B}}\frac{1}{n}\|\tilde{\mathbf{g}}_i\|
$$

$$
\leq \sum_{\mathbf{A}}\|\frac{m-n}{nm}\mathbf{g}_i\| + \sum_{\mathbf{AD}}\|\frac{1}{m}\mathbf{g}_i\| + \sum_{\mathbf{AR}}\|\frac{1}{m}\mathbf{g}_i\| + \sum_{\mathbf{B}}\frac{1}{n}\|\tilde{\mathbf{g}}_i\|
$$

For individual gradient, according to the label corruption gradient definition in problem 2, assuming the $\|\mathbf{W}\|_{op} \leq C$, we have $\|\mathbf{g}_i\| \leq \|\alpha_i\|\|\mathbf{W}_i\|_{op} \leq C\|\alpha_i\|$. Also, denote $\max_i\|\alpha_i\| = k$, $\max_i\|\delta_i\| = v$, we have $\|\mathbf{g}_i\| \leq Ck$, $\|\tilde{\mathbf{g}}_i\| \leq Cv$.

$$
\|\mu(\mathbf{G}) - \mu(\mathbf{N})\| \leq Cx\frac{m-n}{nm}k + C(n-x)\frac{1}{m}k + C(m-n)\frac{1}{m}k + C(n-x)\frac{1}{n}v
$$

Note the above upper bound holds for any $x$, thus, we would like to get the minimum of the upper bound respect to $x$. Rearrange the term, we have

$$
\|\mu(\mathbf{G}) - \mu(\mathbf{N})\| \leq Cx(\frac{m-n}{nm} - \frac{1}{m})k + Cn\frac{1}{m}k + C(m-n)\frac{1}{m}k + C(n-x)\frac{1}{n}v
$$

$$
= C\frac{1}{m}(\frac{2\epsilon-1}{1-\epsilon})xk + Ck + Cv - \frac{1}{n}Cxv
$$

$$
= Cx\left(\frac{k(2\epsilon-1)}{m(1-\epsilon)} - \frac{v}{n}\right) + Ck + Cv
$$

$$
= Cx\left(\frac{k(2\epsilon-1)-v}{m(1-\epsilon)}\right) + Ck + Cv
$$

Since when $\epsilon < 0.5$, $\dfrac{k(2\epsilon-1)-v}{m(1-\epsilon)} < 0$, we knew that $x$ should be as small as possible to continue the bound. According to our algorithm, we knew $n - m\epsilon = m(1-\epsilon) - m\epsilon = (1-2\epsilon)m \leq x \leq n = (1-\epsilon)m$. Then, substitute $x = (1-2\epsilon)m$, we have

$$
\|\mu(\mathbf{G}) - \mu(\mathbf{N})\| \leq Ck(1-2\epsilon)\frac{2\epsilon-1}{1-\epsilon} + Ck + Cv - Cv\frac{1-2\epsilon}{1-\epsilon}
$$

$$
= Ck\frac{3\epsilon-4\epsilon^2}{1-\epsilon} + Cv\frac{\epsilon}{1-\epsilon}
$$

## A.11 PROOF OF THEOREM 3

According to algorithm3, we could guarantee that $v \leq k$, then, we will have:

$$
\begin{aligned}
\|\mu(\mathbf{G}) - \mu(\mathbf{N})\| &\leq Ck\frac{3\epsilon - 4\epsilon^2}{1 - \epsilon} + Cv\frac{\epsilon}{1 - \epsilon} \\
&\leq Ck\frac{4\epsilon - 4\epsilon^2}{1 - \epsilon} \\
&= 4\epsilon Ck \\
&= \mathcal{O}(\epsilon\sqrt{q})(\text{C is constant, k is the norm of } q\text{-dimensional vector})
\end{aligned}
$$

## A.12 COMPARISON BETWEEN SORTING LOSS LAYER GRADIENT NORM AND SORTING THE LOSS VALUE

Assume we have a $d$ class label $\mathbf{y} \in \mathcal{R}^d$, where $y_k = 1, y_i = 0, i \neq k$. We have two prediction $\mathbf{p} \in \mathcal{R}^d, \mathbf{q} \in \mathcal{R}^d$.

Assume we have a $d$ class label $\mathbf{y} \in \mathbb{R}^d$, where $y_k = 1, y_i = 0, i \neq k$. With little abuse of notation, suppose we have two prediction $\mathbf{p} \in \mathbb{R}^d, \mathbf{q} \in \mathbb{R}^d$. Without loss of generality, we could assume that $\mathbf{p}_1$ has smaller cross entropy loss, which indicates $\mathbf{p}_k \geq \mathbf{q}_k$

For MSE, assume we have opposite result

$$
\begin{aligned}
&\|\mathbf{p} - \mathbf{y}\|^2 \geq \|\mathbf{q} - \mathbf{y}\|^2 \\
\Rightarrow &\sum_{i \neq k} p_i^2 + (1 - p_k)^2 \geq \sum_{i \neq k} q_i^2 + (1 - q_k)^2
\end{aligned} \tag{2}
$$

For each $p_i, i \neq k$, We have

$$
Var(p_i) = E(p_i^2) - E(p_i)^2 = \frac{1}{d - 1}\sum_{i \neq k} p_i^2 - \frac{1}{(d - 1)^2}(1 - p_k)^2 \tag{3}
$$

Then

$$
\begin{aligned}
&\sum_{i \neq k} p_i^2 + (1 - p_k)^2 \geq \sum_{i \neq k} q_i^2 + (1 - q_k)^2 \\
\Rightarrow &Var_{i \neq k}(\mathbf{p}_i) + \frac{d}{(d - 1)^2}(1 - p_k)^2 \geq Var_{i \neq k}(\mathbf{q}_i) + \frac{d}{(d - 1)^2}(1 - q_k)^2 \\
\Rightarrow &Var_{i \neq k}(\mathbf{p}_i) - Var_{i \neq k}(\mathbf{q}_i) \geq \frac{d}{(d - 1)^2}\left((1 - q_k)^2 - (1 - p_k)^2\right) \\
\Rightarrow &Var_{i \neq k}(\mathbf{p}_i) - Var_{i \neq k}(\mathbf{q}_i) \geq \frac{d}{(d - 1)^2}\left((p_k - q_k)(2 - p_k - q_k)\right)
\end{aligned} \tag{4}
$$

