# OpenReview forum: "Provable Robust Learning for Deep Neural Networks under Agnostic Corrupted Supervision"
_ICLR.cc/2021/Conference — Reject_

### Official Review · AnonReviewer3 · 2020-10-27
**Simplistic assumptions, misses important prior work**

**Rating:** 3
**Confidence:** 5

**Review:**

# Summary

The papers studies the problem of robust machine learning, where the labels of the a fraction of samples are arbitrarily corrupted. The paper proposes an algorithm to tackle this problem and evaluates it on a standard datasets.

# Positives

The paper studies an important problem prevalent in modern machine learning, and proposes two algorithms to solve these problems. The experiments suggest that the proposed algorithm is better than the baselines.

# Negatives

The paper does not cite highly relevant papers, overclaims its results, and the theoretical results in this paper are immediate. Moreover, the paper is not well-written. More details are given below:
+ Page 1: "Instead of developing an accurate criterion for detection corrupted samples, we adopt a novel perspective and focus on limiting the collective impact of corrupted samples during the learning process through robust mean estimation of gradients."
+ This is not a novel perspective and has been known in robust machine learning community for some time [1,2]. These papers have the same underlying idea, but they are not discussed in this paper. [1] is only briefly mentioned in Remark 2, but the comparison is not fair. The results in [1] hold under fairly general conditions, where the results in this paper require the gradient to be uniformly bounded, which makes the problem significantly simple.
+ Theorem 2 is a trivial result, well-known in field.  Moreover, the way it is presented is misleading and confusing. The error would depend on the quantile of  norms in G, which has been hidden under the O(.) notation. The proof is also missing from the paper.
+ Assumption 1, i.e., Lipschitz continuity of the loss function is very restrictive, which is not satisfied by popular choices of loss function. This assumption trivializes the problem and restricts its applicability.
+ In the same vein, Theorem 3 assumes unrealistic assumptions. The assumption that $||W||_{op} \leq C$ is very restrictive and does not hold for usual learning tasks.	This assumption in a sense is restricting that the covariates x in $R^d$ have bounded norms, whereas the norm of a typical vector in $R^d$ increases as $\sqrt{d}$.

# Score
I propose to reject this paper. Prior work ([1,2]) has studied this problem in a much greater generality, which are not discussed in this work. The assumptions in the present work are severely restrictive.
## Other major comments:
+ Robust linear regression, with arbitrary corruptions in responses, has been extensively studied in the literature but they have not been cited.  For example, see [3,4]. In particular, the least trimmed squares is an algorithm that removes outliers based on loss values, and comes with a theoretical guarantee via an alternating minimization algorithm [3,4].
+ Theorem 1 is a folklore, and this should be reflected in main text. Currently, this information is only given in Appendix.
+ The paper is not well written:
 1.Proof of Theorem 2 is missing.
 2. $O(.)$ notation hides the dependence on the important quantity in the papers.
 3. Important notations have not been defined in the paper.
 4. Abbreviations should not be used, for example, Thm., Algo., Asm., etc.
 5. There are numerous typos and grammatical errors. For example, "has a remarkably impact".

## Relevant papers
1. Diakonikolas, I., G. Kamath, D. M. Kane, J. Li, J. Steinhardt, and A. Stewart. “Sever: A Robust Meta-Algorithm for Stochastic Optimization.” In Proceedings of the 36th International Conference on Machine Learning, ICML 2019, 97:1596–1606. Proceedings of Machine Learning Research. PMLR, 2019. http://proceedings.mlr.press/v97/diakonikolas19a.html.
2. Prasad, A., A. S. Suggala, S. Balakrishnan, and P. Ravikumar. “Robust Estimation via Robust Gradient Estimation.” Journal of the Royal Statistical Society: Series B (Statistical Methodology) 82, no. 3 (July 2020): 601–27. https://doi.org/10.1111/rssb.12364.
3. Bhatia, K., P. Jain, P. Kamalaruban, and P. Kar. “Consistent Robust Regression.” In Advances in Neural Information Processing Systems 30, NeurIPS 2017, 2110–2119, 2017. http://papers.nips.cc/paper/6806-consistent-robust-regression.
4. Bhatia, K., P. Jain, and P. Kar. “Robust Regression via Hard Thresholding.” In Advances in Neural Information Processing Systems 28, NeurIPS 2015, 721–729, 2015. http://papers.nips.cc/paper/6010-robust-regression-via-hard-thresholding.

---

> ### Author Response · Authors · 2020-11-16
> **Response to Reviewer 3**
>
> We thank the reviewer for his/her review and comments to improve the paper. Our point to point response are listed below:
>
> 1) Comparison of Diakonikolas et. al. 2019 and other robust linear regression models.
>
> We fully recognize that Diakonikolas et. al. 2019 achieves better (and very likely optimal) theoretical results in general corruption (i.e. corruption on both X and Y). Thus, we do not claim Theorem 2 as our contribution since the error rate is worse compared to Diakonikolas et. al. 2019 (see three paragraphs below corollary 1, we clearly mentioned that the bound is not dimension-free without Lipschitz assumptions).
>
> Compared with Diakonikolas 19, our paper has two contributions.
>
> a). By assuming the corruption comes from the label (we admit that this is quite strong compared to the general corruption setting), we could get a good error rate.
> b). Our algorithm can be scaled to deep neural networks.
>
> The main reason that Diakonikolas 19 failed at DNN is the space complexity.
> Diakonikolas 19 requires performing SVD on the n by d matrix, where n is the total sample size and d is the number of parameters. Suppose we are using DNN in MNIST data, then n will be around 70000, and d could be millions. In other words, to get the gradient matrix, it requires the space of n-copy of our neural network, which is far beyond the current best GPU memory limitation. In Diakonikolas et. al. 2019, the experiments are performed in ridge regression and SVM, where the number of parameters is usually small compared to DNN. That is why we cannot compare it with Diakonikolas 19.
>
> We noticed that there is a huge gap between theoretical communities and other communities. Currently, the SOTA results for learning under noisy labels are all achieved by deep neural networks. However, few of them have provable guarantees. In the theoretical community, starting from the breakthrough of robust mean estimation, many studies achieved very nice results in terms of robustness. However, those nice algorithms themselves cannot be directly applied to DNN, and many of them study linear/convex models (references 2, 3, and 4 provided by the reviewer). It is unfair to require similar theoretical results for DNN as simple linear models.
>
> 2) Theorem 2 is well-known
>
> We agree that Theorem 2 is well known in the robust mean estimation community. We did not state that as our contribution, and we clearly mentioned in the paper that in general, Theorem 2 is a dimensional dependent result, and we proposed a better algorithm when we assume the label comes from the supervision part, which is our study goal and main contribution.
>
> We did not use the quantile way to present the results and we simplified the results by directly using the 100-quantile. We agree that using quantile is more accurate and we will change the presentation of theorem 2 by introducing the quantile function.
>
> 3). The Lipschitz continuous assumption is strong
>
> The Lipschitz continuous assumption is widely used to study non-convex stochastic optimization. In some literature, stronger assumptions (i.e. convex) are used to study the problem.
>
> Secondly, we agree that this assumption is strong in the robust mean estimation community since it provides a uniform gradient upper bound. We clearly mentioned this limitation below corollary 1.
>
> 4). Whether assume operator norm is bounded is strong
>
> Compared with the uniform gradient upper bound, we only require the largest eigenvalue of the covariance matrix for the gradient matrix to be bounded. Note in [1], the filtering method for robust mean estimation tries to remove/reweight the data point until the spectral norm of the covariance matrix is below some threshold, which indicates that the operator norm of the clean individual matrix is well-bounded. According to our best knowledge, performing robust mean estimation usually comes from the assumption that good data is well-concentrated. Compared to the uniform gradient bound in Theorem 2, assuming the maximum singular value of the gradient matrix is well-bounded is weaker.
>
> 5). Robust linear regression, and trimmed loss baseline
>
> We admit that robust linear regression is long studied, and many promising theoretical results are shown. However, in this paper, we are studying the general non-convex setting, especially for deep neural networks. According to our best knowledge, for deep neural network regression, few of them have robustness guarantees. In the experiment part, we also compared the method that drops data by its loss value (SPL/Trimmed loss), and our results perform much better.
>
>
> [1] Diakonikolas, I., Kamath, G., Kane, D. M., Li, J., Moitra, A., & Stewart, A. (2016, October). Robust Estimators in High Dimensions without the Computational Intractability. In 2016 IEEE 57th Annual Symposium on Foundations of Computer Science (FOCS) (pp. 655-664).

---

> > ### Comment · AnonReviewer3 · 2020-11-23
> > **Missing peer review and simplistic assumptions**
> >
> > I will keep this comment short. I share the sentiments of other reviewers that paper misses many highly related works and makes simplistic assumptions which makes the estimation problem trivial. I stand by my original score and recommend rejection. As the current theoretical results in the paper are either immediate or well-known, perhaps the paper can focus on extensive experimental evaluation of different approaches in a resubmission.
> >
> > **PS**  *"We fully recognize that Diakonikolas et. al. 2019 achieves better (and very likely optimal) theoretical results in general corruption"*.
> >
> > The original version barely mentioned  Diakonikolas et. al. 2019, Prasad et al., and other related works. Even after multiple reviewers have pointed out previous works, the paper still claims robust estimation of gradients to be a novel perspective on page 1. It gives an impression that paper is over claiming its results.

---

> > > ### Author Response · Authors · 2020-11-24
> > > **Further Response**
> > >
> > > Thank you for the feedback. And we have a few remarks to briefly communicate before the closure of the interaction:
> > >
> > > 1 Significance
> > >
> > > The related literature as pointed out by reviewer is fundamentally different from our work and NOT directly comparable because ONLY our approach can be applied to deep learning, that covered a majority of research nowadays.
> > >
> > > As we have pointed out previously: (a) Diakonikolas et. al. 19 cannot be applied to DNN due to its high space complexity. (b) Prasad et al. considered only convex loss. (c) the two papers by Bhatia et. al. are designed for linear regression instead of non-linear models.
> > >
> > > Besides, we DID discuss Diakonikolas et. al. 19 in remarks of the original submission. However, due to page limitation, we were not able to expand the related work section. In the rebuttal session, we added two more paragraphs to discuss the related work as suggested.
> > >
> > > 2 Assumptions
> > >
> > > We have three assumptions in the paper that are either commonly used in recent literature or related to a standard problem setting. We are not clear why the reviewer refer these as “simplified setting”.
> > >
> > > (a) Lipschitz continuity is a very common assumption in machine learning, as also used in Diakonikolas et. al. 19 (see proposition 2.2 or proposition B.5)
> > >
> > > (b) Bounded maximum singular value is also used by Diakonikolas et. al. 19 (see Theorem 2.1 or the first sentence in section “More general assumptions” in Diakonikolas et. al. 19)
> > >
> > > (c) Corruptions happen on labels is an important problem setting widely investigated in both academia and industry (a selected list is https://github.com/subeeshvasu/Awesome-Learning-with-Label-Noise), and should not be considered a simplified assumption.
> > >
> > > 3 Empirical study
> > >
> > > We appreciate the suggestion from the reviewer for extending the experiments for resubmission, but we would like to point out that our experiments are much more extensive than the literature referred by the reviewer. It will be great if the reviewer could elaborate further on this for us to improve.

---

### Official Review · AnonReviewer2 · 2020-10-28
**A robust noisy label learning algorithm with meaningful insight but lack theoretical proofs**

**Rating:** 5
**Confidence:** 3

**Review:**

pros
1. The authors provide an insight that in noisy label learning, if the corrupted gradient is not far from the true one, then the learning algorithm could converge a sub-optimal result.
2. Detailed experiments show the empirical evidence of the proposed algorithm over different kinds of label noise.

cons
1. The authors propose a method that only keeping the data with a small gradient norm in the training process to resist label noise. However, they do not verify that such a design is motivated by their theoretical results. Some important proofs for their key results are missing, e.g., Theorem 2 and Theorem 3, making this paper not self-contain.
1. Many symbols are not well-defined mathematically.


This paper proposes a robust algorithm for noisy label learning. By keeping the data with a small gradient norm in the training process, the proposed algorithm could resist the label noise. Instead of making assumptions on the label corruption, the authors assume that the difference between the clean mini-batch gradient and the corrupted mini-batch gradient is bounded. Thus the proposed method could converge to the $\epsilon$-optimal results. By dropping the data with a large gradient norm, the estimated gradient mean will not be far from the true one. The theoretical results make sense, but there lack detailed proofs to make this paper self-contain, e.g., for Theorem 2 and Theorem 3. The empirical studies on several datasets show the robustness of the proposed algorithm over different kinds of label noise.

---

> ### Author Response · Authors · 2020-11-16
> **Response to Reviewer 2**
>
> We thank the reviewer for his/her review and comments to improve the paper. Our point to point response are listed below:
>
> 1) The authors do not verify that such a design is motivated by their theoretical results.
>
> Our method is a filtering based method. Filtering data out during training is a common approach against noisy labels, and widely used in many papers.
>
> The motivation for using the loss layer gradient norm to filter data is from lemma 3. Lemma 3 states that randomized filtering algorithm results are affected by the loss-layer gradient norm. Thus, we decided to use the loss-layer gradient norm as the dropping criteria, which could decrease the upper bound of gradient estimation error.
>
>
> 2) Missing proof for Theorem2 and Theorem3
>
> Sorry for not highlighting the proof of theorem 3. Theorem 3 is directly from Lemma 3 and Corollary 1. We provided the proof of lemma 3, and the last part of this proof actually gives proof of theorem 3. We now split the proof of lemma 3 and proof of theorem 3 in the appendix.
>
> We are sorry for missing proof for Theorem 2. Theorem2 is classical results in the robust mean estimation, and this is not our main contribution. We added the proof in the appendix already.

---

### Official Review · AnonReviewer1 · 2020-10-29
**Novelty of the results is questionable; combining robust gradient learning with collaborative learning lacks strong motivation.**

**Rating:** 4
**Confidence:** 5

**Review:**


1. The related work section misses MANY related results on corrupted data and robust mean estimation.
2. The related work section forget to mention existing theoretical results that apply robust mean estimation for robust gradient calculation.
3. The related work section does not provide an accurate overview of existing results. For example, "the algorithms themselves are NP-hard" is not the correct statement -- NP-hard describes the hardness of a problem, not an algorithm.
4. Collaborative learning methods seem to have no solid theoretical understanding and it is unclear why the proposed algorithm build on top of it.
5. Regarding the novelty of the theorems: Theorem 1 studies convergence of biased gradient, which is another known research topic and has been studied before, but the authors have not discuss/compare their results with existing ones and the novelty may be overclaimed. Theorem 3 is for robustness guarantee with corruption only in the supervision, and existing results have shown O(\epsilon) guarantee (for linear regression and its variants).
6. I have not listed all the missing literature (I believe they are easy to find after a careful literature review), but I can add comment later if needed.

---

> ### Author Response · Authors · 2020-11-16
> **Response to reviewer 1**
>
> We thank the reviewer for his/her review and comments to improve the paper. Our point to point response are listed below:
>
> 1) Missing Related Work
>
> We thank the reviewer for pointing out missing related work. The page is limited and we tried our best to include important related work. We added more literature in the background section.
>
> 2) NP-hard description.
>
> Thank you for pointing out the mistake. We change it to the algorithm is not a polynomial-time algorithm instead of saying it is NP-hard.
>
> 3) Why build on top of the collaborative learning method, which is not well theoretical motivated.
>
> The collaborative learning method is from [1]. Indeed, this paper does not have a good theoretical motivation, but it achieves the SOTA performance, and many other paper studies noisy labels are built on the top of the co-teaching framework [2, 3, 4, 5] (more papers can be found in the third-party repo).
>
> 4) Theorem 1 is widely studied.
>
> Theorem 1/Theorem 4 is to motivate the robust gradient estimation. In the appendix, we clearly mentioned that Theorem 4 is a standard result and gives reference with similar results, and we did not claim that as our contribution.
>
> 5) Question about Theorem 3
>
> Theorem 3 is for corrupted supervision, and linear regression methods already solve this problem with O(eps) error bound.
> Studying learning under corrupted supervision is the goal of this paper, and the reviewer mentioned that existing results have shown O(\epsilon) guarantee for linear regression and its variants. However, in our work, we study the DNN based regression and classification method, and it is unfair to compare with theoretical results in linear regression settings.
>
> [1] Han, B., Yao, Q., Yu, X., Niu, G., Xu, M., Hu, W., ... & Sugiyama, M. (2018). Co-teaching: Robust training of deep neural networks with extremely noisy labels. In Advances in neural information processing systems (pp. 8527-8537).
> [2] Song, H., Kim, M. and Lee, J.G., 2019, May. Selfie: Refurbishing unclean samples for robust deep learning. In International Conference on Machine Learning (pp. 5907-5915).
> [3] Xingrui Yu, Bo Han, Jiangchao Yao, Gang Niu, Ivor Tsang, and Masashi Sugiyama., 2019. How does disagreement help generalization against label corruption? In International Conference on Machine Learning, (pp. 7164-7173).
> [4] Wei, H., Feng, L., Chen, X. and An, B., 2020. Combating noisy labels by agreement: A joint training method with co-regularization. In Proceedings of the IEEE/CVF Conference on Computer Vision and Pattern Recognition (pp. 13726-13735).
> [5] Wang, X., Wang, S., Wang, J., Shi, H. and Mei, T., 2019. Co-mining: Deep face recognition with noisy labels. In Proceedings of the IEEE international conference on computer vision (pp. 9358-9367).

---

### Official Review · AnonReviewer4 · 2020-10-29

**Rating:** 4
**Confidence:** 4

**Review:**

In this paper, the authors studied the problem of training neural networks under data poisoning, i.e., when a small fraction of the training data is corrupted by the adversary. They considered two data corruption settings, one allows both the data x and supervision y to be corrupted, which is called general corruption,  and one with only supervision y corrupted. Their first algorithm, which removes the datapoints whose gradient norm is large when computing the average gradient, applies to the general supervision setting. They showed their algorithm has eps\sqrt(d) error or eps*L error, which can be quite large for high-dimensional and deep neural nets learning settings. Their second algorithm applies to the setting where only supervision y is corrupted, and the algorithm works by removing the datapoints whose output layer gradient is large. Assuming the clean data has bounded gradient, and the dimension of y is p, their algorithm achieves error eps*sqrt(p).

Weakness:
1.The authors claimed that compared to Diakonikolas 19, they improved the error from eps to sqrt(eps). However, the eps result relies on the fact that the gradient of good data has bounded norm, and I believe in that setting Diakonikolas 19 also achieves eps error.
2. In paragraphs close to Lemma 1 and Lemma 3, the authors mentioned a randomized filtering algorithm, and proved Lemma 1 Lemma 3 for the algorithm. However, I can’t find the mentioned randomized filtering algorithm in the paper.
3. Theorem 1 and Theorem 4 have no formal proof.
4. Theorem 2 has no proof.
5. In the experiment section, there is no comparison to other state-of-the-art algorithms, for example Diakonikolas 19.

Overall, I think the theoretical result in the paper is incomplete, and the experimental evaluation is insufficient.

---

> ### Author Response · Authors · 2020-11-16
> **Response to Reviewer 4**
>
> We thank the reviewer for his/her review and comments to improve the paper. Our point to point response are listed below:
>
> 1. The authors claimed that compared to Diakonikolas 19, they improved the error from eps to sqrt(eps). However, the eps result relies on the fact that the gradient of good data has bounded norm, and I believe in that setting Diakonikolas 19 also achieves eps error.
>
> Firstly, we did not see the O(eps) error rate results in Diakonikolas et. al. 2019, and in the section “Stronger robustness to outliers” of Diakonikolas et. al. 2019, it said that it remains to study whether it is possible to achieve O(eps) error without strong assumptions. We would very much appreciate it if you could tell us where the paper states the O(eps) results?
>
> Secondly, in our paper, the gradient of good data has bounded norm is from the individual Lipschitz continuous, and we admitted that the error rate we achieved is dimension dependent without the individual Lipschitz continuous(see three paragraphs below Corollary 1). However, by assuming the corruption only happens on the label part, we find that we could achieve dimension-free results, and in Theorem 3, we only assume the largest singular value of the gradient matrix is bounded.
>
> 2. In paragraphs close to Lemma 1 and Lemma 3, the authors mentioned a randomized filtering algorithm and proved Lemma 1 Lemma 3 for the algorithm. However, I can’t find the mentioned randomized filtering algorithm in the paper.
>
> Sorry for making you confused about the randomized filtering algorithm, this randomized filtering algorithm is not our final algorithm. The purpose of studying the randomized filtering algorithm is to study what affects the quality of the final solution. In lemma 3, we find the randomized filtering results are highly impacted by the loss-layer gradient norm (i.e. v in lemma 3), and that motivates algorithm 3, which is dropping the data by checking the loss-layer gradient norm.
>
> 3. Theorem 1 and Theorem 4 have no proof, Theorem has no proof.
> Sorry for missing the formal proof. We added formal proof in the appendix. The reason that we did not add formal proof is Theorem/Theorem4 are actually standard results, and widely studied by other researchers. We do not claim that as our contribution. In the appendix, we list some literature that states very similar results.
>
> For Theorem 2, this theorem is also a classical result in the robust mean estimation community, and we did not provide the proofs. We will add detailed proof later.
>
> 4. Did not compare with Diakonikolas et. al. 2019 in experiment.
>
> In our paper, we never claimed that our paper is theoretically better than Diakonikolas et. al. 2019 in their setting (corruption in both x and y). However, several things make Diakonikolas et. al. 2019 cannot be applied in deep neural networks, and DNN based noisy label learning archives currently SOTA results (empirically). Compared to Diakonikolas et. al. 2019, we have two contributions:
> a). By assuming the corruption comes from the label (we admit that this is quite strong compared to the general corruption setting), we could get a better error rate.
> b). Our algorithm can be scaled to deep neural networks while Diakonikolas et. al. 2019 cannot. That is why in experiments we cannot compare with Diakonikolas et. al. 2019.
>
> The main reason that Diakonikolas 19 failed at DNN is the space complexity.
> Diakonikolas 19 requires performing SVD on the n by d matrix, where n is the total sample size and d is the number of parameters. Suppose we are using DNN in MNIST data, then n will be around 70000, and d could be millions. In other words, to get the gradient matrix, it requires the space of n-copy of our neural network, which is far beyond the current best GPU memory limitation (we also mentioned that in remark 2). In Diakonikolas 19, the experiments are performed in ridge regression and SVM, where the number of parameters is usually smaller compared to DNN. That is why we can not compare it with Diakonikolas et. al. 2019 in the experiment.
>
> Although our algorithm has several strong assumptions (corruptions on the label) and compared to Diakonikolas et. al. 19, the theoretical results are not as general as it is, our algorithm is much more practical in terms of deep neural networks. Also, corruption on the label is also a research area, which aims to provide better results if we assume the corruption happens only from the supervision part (https://github.com/subeeshvasu/Awesome-Learning-with-Label-Noise). Our baseline (co-teaching) [1] is actually one of the SOTA in terms of noisy label learning.
>
> We add a paragraph in the paper that discusses the relation of Diakonikolas et. al. 2019.
>
> [1] Han, B., Yao, Q., Yu, X., Niu, G., Xu, M., Hu, W., ... & Sugiyama, M. (2018). Co-teaching: Robust training of deep neural networks with extremely noisy labels. In Advances in neural information processing systems (pp. 8527-8537).

---

### Author Response · Authors · 2020-11-16
**General Response for Reviewer's Questions**

We thank the reviewers for all comments to improve the paper. We provide the point to point response for each reviewer, but we would like to first clarify several general questions raised by reviewers:

1). In our paper, we did NOT claim that we provide better theoretical results than (Diakonikolas et. al. 19) in a general agnostic corruption setting, and our study scenario is robust learning under LABEL corruption, which is a different problem. More importantly, when we assume the corruptions only come from labels, our algorithm (algorithm 3) achieves better results.

Although we mentioned algorithm 2 (for general corruption) in the paper, we stated that it has several drawbacks (i.e. not dimension-free, not efficient), and algorithm 2 is NOT our main algorithm since our goal is to study the label corruption instead of general corruptions. We claim that when we assume that only labels have corruption, we could improve algorithm 2. By using algorithm3, we can achieve better theoretical results, which hold for complex models like deep neural networks (DNNs). According to our best knowledge, most existing DNN SOTA methods do not provide any theory on agnostic label corruption setting (in both regression and classification). We provided a third-party paper collection repo here (https://github.com/subeeshvasu/Awesome-Learning-with-Label-Noise) to make the reviewers convenient to see the community difference between robust mean estimation/robust optimization and learning with label noise (mostly using DNN).

2). We also would like to mention that although Diakonikolas 19 et. al. achieves very nice theoretical results, unfortunately, this method cannot be applied to DNN giving the current best hardware configuration, and DNN based models are currently state-of-the-art methods for noisy label learning problems (at least in empirical performance). Diakonikolas 19 et. al. uses dimension-free robust mean estimation breakthroughs to design the learning algorithm, while we notice that most robust mean estimation relies on filtering out data by computing the score of projection to maximum singular vector. For example, in Diakonikolas 19 et. al., it requires to compute SVD on n by d individual gradient matrix, where n is the sample size and d is the number of parameters. This method works well for small datasets and small models since both n and d is small enough for current memory limitation. However, for deep learning methods, this matrix size is far beyond current GPU memory capability. That could be the potential reason why in Diakonikolas 19, only ridge regression and SVM results for small data are shown (we are not saying that they should provide DNN results). In our experiment, our n is 60000 and d is in the magnitude of millions (network parameters). It is obviously impractical to store 60000 copies of neural networks in a single GPU card. In contrast, in our algorithm (algorithm 3), we do not need to store the full gradient matrix. By only considering the loss-layer gradient norm, we could easily extend our algorithm to DNN, and we showed that this simple strategy works well in both theory and challenging empirical tasks.

3). We also would like to clarify that many reviewers mentioned that the theory of robust regression is long studied. However, most of them are very specific for linear regression or its variants. According to our best knowledge, we did not see any DNN based robust regression work with theoretical guarantees in an agnostic label corruption setting. We provide a third-party repo here (most of them are DNN method)(https://github.com/subeeshvasu/Awesome-Learning-with-Label-Noise), and in this repo, we did not find any noisy label for regression.

We will add those robust linear regression methods to the references, and highlight those work in the introduction part. However, we think that it is unfair to require a deep method having the same promising theoretical results as linear or convex methods.

In order to address the reviewer’s concern, we add the following content in our paper:
(a) we add a paragraph in related work to include more literature in robust optimization.
(b) we add a section to describe the comparison with Diakonikolas 19 et. al.
(c) we add proof for theorems in the appendix.
(d) Below theorem 2, we emphasize more the disadvantage of theorem 2.
(e) we change our paper title to Provable Robust Learning for Deep Neural Networks under Agnostic Corrupted Supervision to highlight that our algorithm should be considered/compared to other DNN methods for learning under noisy labels.

All changed places are highlighted in red.

---

### Author Response · Authors · 2020-11-23
**Author Response& Rivision**

Dear reviewers,

Could you please go over our responses and the revision since we can have interactions with you only by this Tuesday (24th)? We have responded to your comments and faithfully reflected them in the revision.

(1) We already mentioned that Diakonikolas et. al. 19 cannot be applied to deep neural networks.

(2) Compared to Diakonikolas et. al. 19, the only strong assumption we made is about corruption happens only on labels, and we provided the 3rd party repo to show that learning under label corruption is widely studied in the DNN community. Thus, the assumption that corruption happens only on labels should not be considered a reason to reject the paper.

(3) We also clearly discussed the relationship between Diakonikolas et. al. 19  in our paper. We clearly stated that Diakonikolas et. al. 19 has better theoretical results in general corruption, but studying general corruption is not our paper's goal.

(4) We also add deep neural networks to the title since all experiments are done in DNN, and our theory holds for deep neural networks. It is unfair to compare our theory to linear models or its variants.

Thanks,
Authors

---

### Decision · Program_Chairs · 2021-01-07
**Final Decision**

**Decision:**

Reject

**Comment:**

The papers studies machine learning tasks in the presence of adversarially corrupt data (during training). In particular, it is assumed that the labels of a small constant fraction of the datapoints are arbitrarily corrupted.  The paper proposes a natural method to solve this problem and evaluates it on various datasets. As pointed out by the reviewers, the theoretical contributions of this paper are subsumed by a number of prior works (which were not initially cited). The experimental results of the paper are interesting. However, the method proposed  and evaluated is not particularly novel. In my opinion, the problems studied in this submission are important (in particular, the memory/space consideration in the context of robustness). However, this work still needs work and is not ready for publication.